# RBP-J regulates homeostasis and function of circulating Ly6C$^{lo}$ monocytes

Tiantian Kou[1,2,3†], Lan Kang[1,3†], Bin Zhang[1,3], Jiaqi Li[1], Baohong Zhao[4,5], Wenwen Zeng[1,2,3], Xiaoyu Hu[1,2,3]*

[1]Institute for Immunology and School of Medicine, Tsinghua University, Beijing, China; [2]Tsinghua-Peking Center for Life Sciences, Tsinghua University, Beijing, China; [3]Beijing Key Laboratory for Immunological Research on Chronic Diseases, Beijing, China; [4]Arthritis and Tissue Degeneration Program and the David Z. Rosensweig Genomics Research Center, Hospital for Special Surgery, New York, United States; [5]Department of Medicine, Weill Cornell Medical College, New York, United States

**Abstract** Notch-RBP-J signaling plays an essential role in the maintenance of myeloid homeostasis. However, its role in monocyte cell fate decisions is not fully understood. Here, we showed that conditional deletion of transcription factor RBP-J in myeloid cells resulted in marked accumulation of blood Ly6C$^{lo}$ monocytes that highly expressed chemokine receptor CCR2. Bone marrow transplantation and parabiosis experiments revealed a cell-intrinsic requirement of RBP-J for controlling blood Ly6C$^{lo}$CCR2$^{hi}$ monocytes. RBP-J-deficient Ly6C$^{lo}$ monocytes exhibited enhanced capacity competing with wildtype counterparts in blood circulation. In accordance with alterations of circulating monocytes, RBP-J deficiency led to markedly increased population of lung tissues with Ly6C$^{lo}$ monocytes and CD16.2$^+$ interstitial macrophages. Furthermore, RBP-J deficiency-associated phenotypes could be genetically corrected by further deleting *Ccr2* in myeloid cells. These results demonstrate that RBP-J functions as a crucial regulator of blood Ly6C$^{lo}$ monocytes and thus derived lung-resident myeloid populations, at least in part through regulation of CCR2.

*For correspondence:
xiaoyuhu@tsinghua.edu.cn

†These authors contributed equally.

## eLife assessment

This study presents a **valuable** examination into the role Notch-RBP-J signaling in regulating monocyte subset homeostasis. The data were collected and analyzed using **solid** and validated methodology and can be used as a starting point for exploring the mechanisms involved in RBP-J signaling in non-classical monocytes. The data presented strongly confirm the authors conclusions. However, this article primarily focuses on providing a description, and additional studies are necessary to fully elucidate the mechanisms through which RBP-J deficiency contributes to the specific increase in Ly6C$^{lo}$ monocyte numbers in both the blood and lungs.

## Introduction

Monocytes are integral components of the mononuclear phagocyte system that develop from monocyte precursors in the bone marrow (*Guilliams et al., 2018*). Several functionally and phenotypically distinct subsets of blood monocytes have been defined on the basis of expression of surface markers (*Cros et al., 2010*; *Geissmann et al., 2003*; *Passlick et al., 1989*; *Weber et al., 2000*). In mice, Ly6C$^{hi}$ monocytes (also called inflammatory or classical monocytes) characterized by Ly6C$^{hi}$CCR2$^{hi}$CX3CR1$^{lo}$ exhibit a short half-life and are recruited to tissue and differentiate into macrophages and dendritic cells (*Ginhoux and Jung, 2014*). Ly6C$^{hi}$ monocytes are analogous to CD14$^+$ human monocytes based on gene expression profiling (*Ingersoll et al., 2010*). A second subtype of monocytes is called Ly6C$^{lo}$

monocytes (also termed patrolling monocytes or non-classical monocytes), which express high level of CX3CR1 and low level of CCR2, equivalent to $CD14^{lo}CD16^{+}$ human monocytes. $Ly6C^{lo}$ monocytes are regarded as blood-resident macrophages that patrol blood vessels and scavenge microparticles attached to the endothelium under physiological conditions, and exhibit a long half-life in the steady state (*Auffray et al., 2007*; *Carlin et al., 2013*; *Yona et al., 2013*). $Ly6C^{lo}$ monocytes may exert a protective effect by suppressing atherosclerosis in mice, while high levels of non-classical monocytes have been associated with more advanced vascular dysfunction and oxidative stress in patients with coronary artery disease (*Hamers et al., 2012*; *Hanna et al., 2012*; *Urbanski et al., 2017*). In the inflammatory settings, $Ly6C^{lo}$ monocytes often show anti-inflammatory properties, yet also exhibit a proinflammatory role under certain circumstances (*Amano et al., 2005*; *Brunet et al., 2016*; *Carlin et al., 2013*; *Cros et al., 2010*; *Misharin et al., 2014*; *Nahrendorf et al., 2007*; *Puchner et al., 2018*). In addition, $Ly6C^{lo}$ monocytes have demonstrated protective functions in tumorigenesis, such as engulfing tumor materials to prevent cancer metastasis, activating and recruiting NK cells to the lungs (*Hanna et al., 2015*; *Thomas et al., 2016*). While the functions of $Ly6C^{lo}$ monocytes are complex and sometimes contradictory depending on the animal models and experimental approaches, it is clear that they play a crucial role in both health and disease.

The generation and maintenance of $Ly6C^{lo}$ monocytes are regulated by several transcription factors, including Nr4a1 and CEBPβ (*Hanna et al., 2011*; *Tamura et al., 2017*). The colony-stimulating factor 1 receptor (CSF1R) signaling pathway is important for the generation and survival of $Ly6C^{lo}$ monocytes, and neutralizing antibodies against CSF1R can reduce their numbers (*MacDonald et al., 2010*). Interestingly, recent studies have suggested that different subsets of monocytes may be supported by distinct cellular sources of CSF-1 within bone marrow niches, in which targeted deletion of *Csf1* from sinusoidal endothelial cells selectively reduces $Ly6C^{lo}$ monocytes but not $Ly6C^{hi}$ monocytes (*Emoto et al., 2022*). LAIR1 has been shown to be activated by stromal protein Colec12, and LAIR1 deficiency leads to aberrant proliferation and apoptosis in bone marrow non-classical monocytes (*Keerthivasan et al., 2021*). These findings further underscore the complexity of the mechanisms that regulate $Ly6C^{lo}$ monocytes and suggest that multiple factors are involved in this process.

Recombinant recognition sequence binding protein at the Jκ site (RBP-J; also named CSL) is commonly known as the master nuclear mediator of canonical Notch signaling (*Radtke et al., 2013*). In the absence of Notch intracellular domain, RBP-J associates with co-repressor proteins to repress transcription of downstream target genes. In the immune system, the best studied functions of Notch signaling are its roles in regulating the development of lymphocytes, such as T cells and marginal zone B cells (*Radtke et al., 2013*). Notch signaling also has been reported to regulate the differentiation and function of myeloid cells including granulocyte/monocyte progenitors, osteoclasts, and dendritic cells (*Caton et al., 2007*; *Klinakis et al., 2011*; *Lewis et al., 2011*; *Zhao et al., 2012*). In osteoclasts, RBP-J represses osteoclastogenesis, particularly under inflammatory conditions (*Zhao et al., 2012*). In dendritic cells, Notch-RBP-J signaling controls the maintenance of $CD8^{-}$ DC (*Caton et al., 2007*). In addition, Notch2-RBP-J pathway is essential for the development of $CD8^{-}ESAM^{+}$ DC in the spleen and $CD103^{+}CD11b^{+}$ DC in the lamina propria of the intestine (*Lewis et al., 2011*). However, the role of Notch-RBP-J signaling pathway in regulating monocyte subsets remains elusive. Here, we demonstrated that RBP-J controlled homeostasis of blood $Ly6C^{lo}$ monocytes in a cell-intrinsic manner. RBP-J deficiency led to a drastic increase in blood $Ly6C^{lo}$ monocytes, lung $Ly6C^{lo}$ monocytes, and $CD16.2^{+}$ IM. Our results suggest that RBP-J plays a critical role in regulating the population and characteristics of $Ly6C^{lo}$ monocytes in the blood and lung.

## Results

### RBP-J is essential for the maintenance of blood Ly6Clo monocytes

To investigate the role of RBP-J in monocyte subsets, we utilized mice with RBP-J specific deletion in the myeloid cells (*Rbpj$^{fl/fl}$Lyz2$^{cre/cre}$* mice). Efficient deletion of *Rbpj* in blood monocytes was confirmed by quantitative real-time PCR (qPCR) (*Figure 1—figure supplement 1A*). Flow cytometric analysis revealed that RBP-J-deficient mice had a significant increase in the proportion of $Ly6C^{lo}$ monocytes, but not $Ly6C^{hi}$ monocytes, in blood, compared to age-matched control mice with the genotype *Rbpj$^{+/+}$Lyz2$^{cre/cre}$* (*Figure 1A*). In contrast to circulating monocytes, RBP-J-deficient mice exhibited minimal alterations in the percentages of monocyte subsets in bone marrow (BM) and spleen (*Figure 1B*,

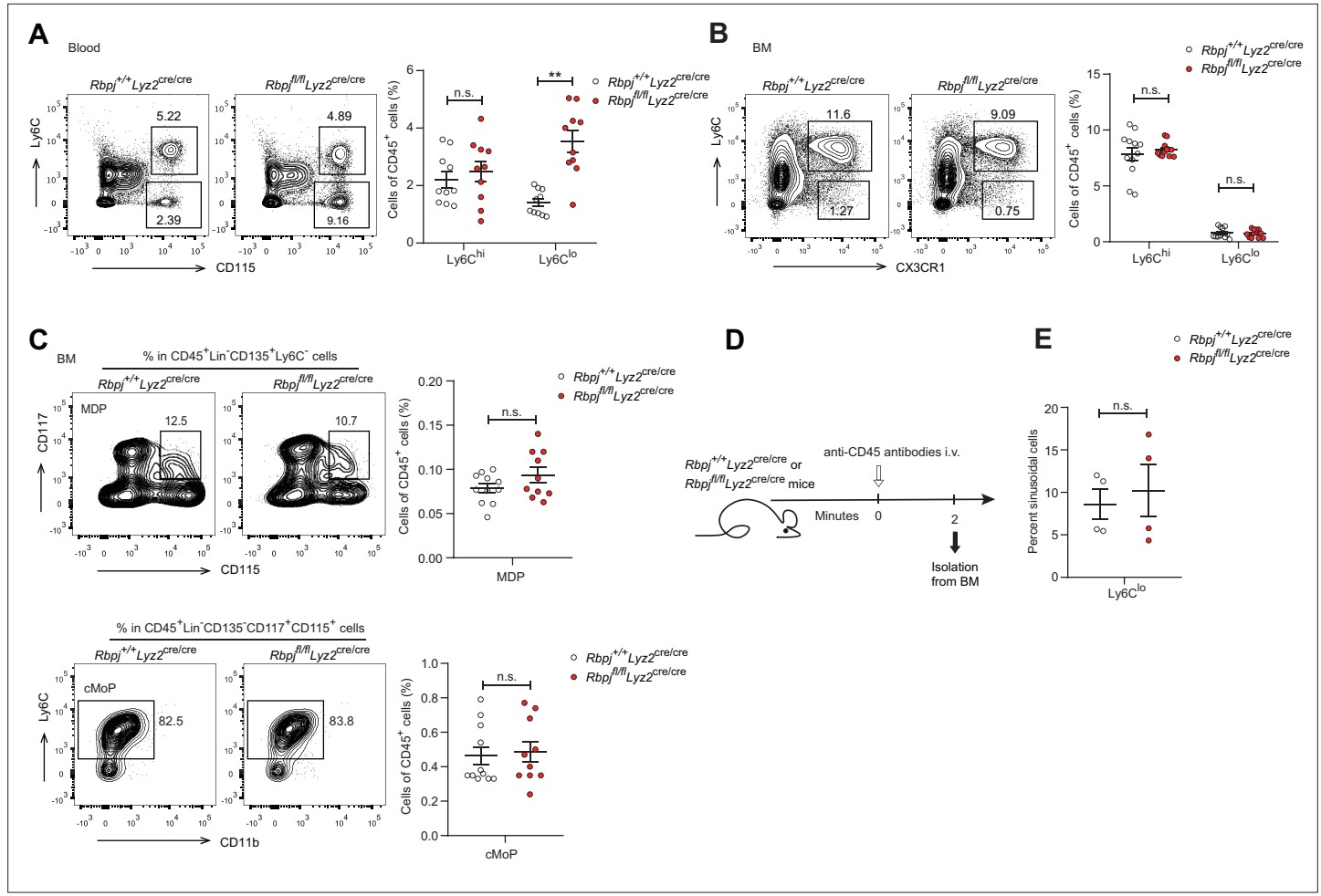

**Figure 1.** RBP-J-deficient mice display more blood Ly6C$^{lo}$ monocytes. (**A**) Blood Ly6C$^{hi}$ and Ly6C$^{lo}$ monocytes in *Rbpj$^{+/+}$Lyz2$^{cre/cre}$* control and *Rbpj$^{fl/fl}$Lyz2$^{cre/cre}$* mice were determined by flow cytometry analyses (FACS). Representative FACS plots (left) and cumulative data of cell ratio (right) are shown. (**B, C**) Representative FACS plots and cumulative data quantitating percentages of bone marrow (BM) monocyte subsets (**B**) and myeloid progenitor cells (**C**) (CD45$^{+}$CD11b$^{+}$Ly6G$^{-}$CD115$^{+}$ Ly6C$^{hi}$ monocyte; CD45$^{+}$CD11b$^{+}$Ly6G$^{-}$CD115$^{+}$Ly6C$^{lo}$ monocyte; MDP, CD45$^{+}$Lin$^{-}$CD117$^{+}$CD115$^{+}$CD135$^{+}$ Ly6C$^{-}$; cMoP, CD45$^{+}$Lin$^{-}$CD11b$^{-}$CD117$^{+}$CD115$^{+}$ CD135$^{-}$Ly6C$^{+}$). Lin: CD3, B220, Ter119, Gr-1 and CD11b. (**D**) Experimental outline for panel (**E**). (**E**) Cumulative data quantitating percentages of sinusoidal monocytes (CD45$^{+}$) within total BM Ly6C$^{lo}$ monocytes. Data are pooled from at least two independent experiments; n ≥ 4 in each group. Data are shown as mean ± SEM; n.s., not significant; **p<0.01 (two-tailed Student's unpaired *t*-test). Each symbol represents an individual mouse.

The online version of this article includes the following source data and figure supplement(s) for figure 1:

**Source data 1.** Data for *Figure 1*.

**Figure supplement 1.** Control and RBP-J-deficient mice show similar neutrophils.

**Figure supplement 1—source data 1.** Data for *Figure 1—figure supplement 1*.

*Figure 1—figure supplement 1C*). Additionally, RBP-J deficiency in myeloid cells did not appear to have an effect on neutrophils (*Figure 1—figure supplement 1B*). Therefore, among circulating myeloid populations, RBP-J selectively controlled the subset of Ly6C$^{lo}$ monocytes.

BM progenitors that give rise to circulating monocytes are monocyte-dendritic cell progenitors (MDPs) and common monocyte progenitors (cMoPs) (*Hanna et al., 2011*; *Hettinger et al., 2013*; *Liu et al., 2019*; *Varol et al., 2007*). Next, we analyzed BM progenitors and found that the percentages of MDPs and cMoPs were equivalent in control and RBP-J-deficient mice (*Figure 1C*). These results in conjunction with the observations of normal BM monocyte populations, suggest that alterations of peripheral blood Ly6C$^{lo}$ monocytes may not originate from BM. Next, we examined whether RBP-J may influence the egress of Ly6C$^{lo}$ monocytes from BM and measured BM exit rate of Ly6C$^{lo}$ monocytes using *in vivo* labeling of sinusoidal cells as previously described (*Figure 1D*; *Debien et al.,*

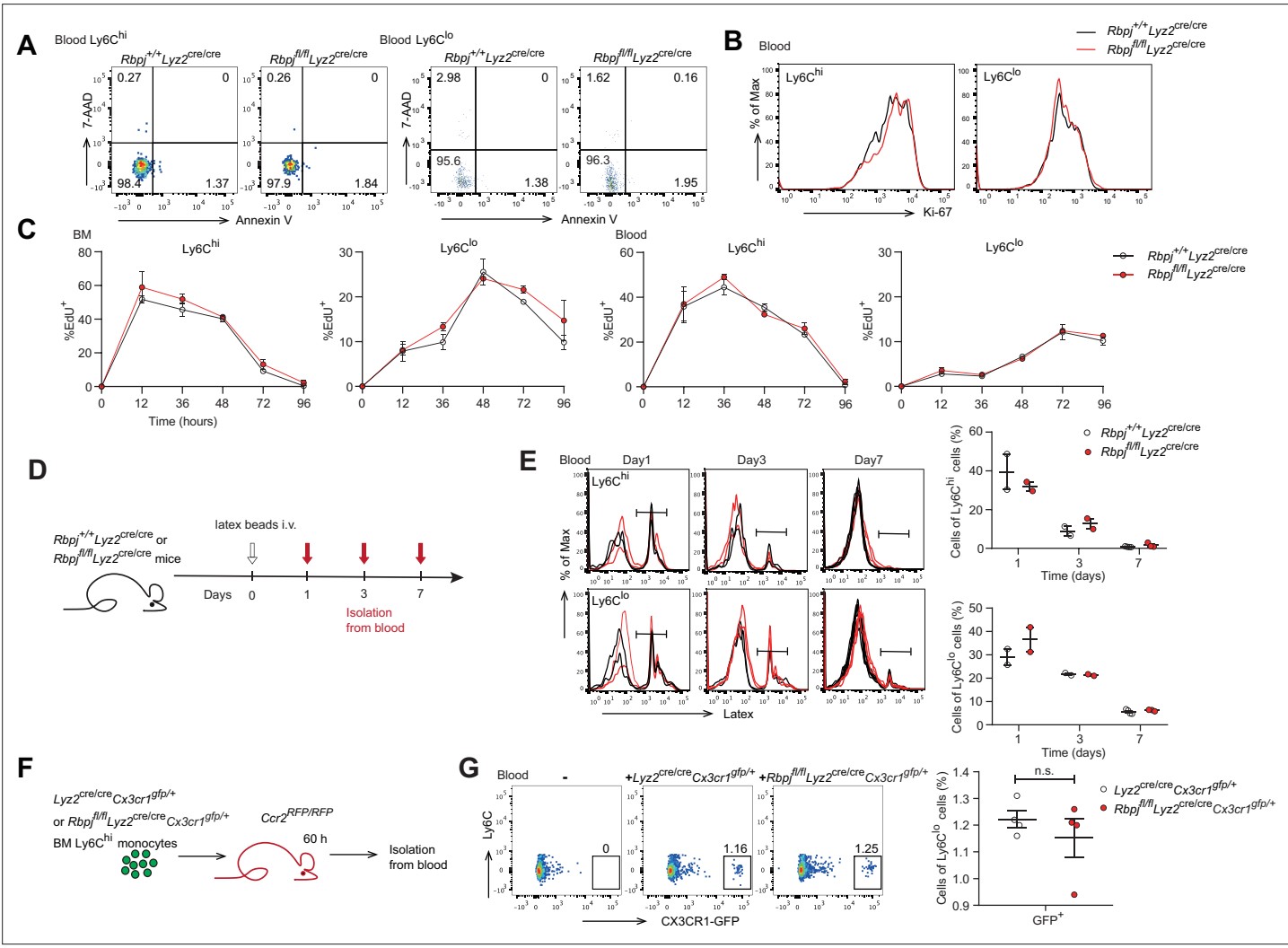

**Figure 2.** Monocyte subsets in RBP-J-deficient mice display normal cell death. (**A**) Representative FACS plots of monocyte subsets in blood stained with 7-amino-actinomycin D (7-AAD) and Annexin V. (**B**) FACS analysis of Ki-67 expression in *Rbpj*[+/+]*Lyz2*[cre/cre] control and *Rbpj*[fl/fl]*Lyz2*[cre/cre] blood monocyte subsets. Black lines represent control mice, and red lines represent RBP-J-deficient mice. (**C**) Analysis of time course of EdU incorporation of monocyte subsets in bone marrow (BM) and blood after a single 1 mg EdU pulsing. The percentages of EdU+ cells among the indicated monocyte subsets are shown. (**D**) Experimental outline for panel (**E**). (**E**) Analysis of time course of latex beads incorporation of monocyte subsets in blood after latex beads injection. The percentages of latex+ cells among the indicated monocyte subsets are shown. (**F**) Cartoon depicting the adoptive transfer. BM GFP+Ly6C[hi] monocytes were sorted from *Lyz2*[cre/cre]*Cx3cr1*[gfp/+] or *Rbpj*[fl/fl]*Lyz2*[cre/cre]*Cx3cr1*[gfp/+] mice and transferred into *Ccr2*[RFP/RFP] recipient mice. Sixty hours after transfer, cell fate was analyzed. (**G**) Representative FACS plots are shown in the left panel, and the frequencies of GFP+Ly6C[lo] monocytes within total Ly6C[lo] monocytes are shown in the right panel. Data are pooled from two independent experiments (**G**); n ≥ 2 in each group (**C, E, G**). Data are shown as mean ± SEM; n.s., not significant; (two-tailed Student's unpaired *t*-test). Each symbol represents an individual mouse (**E, G**).

The online version of this article includes the following source data and figure supplement(s) for figure 2:

**Source data 1.** Data of *Figure 2*.

**Figure supplement 1.** The conversion of Ly6C[hi] monocyte is identical in control and RBP-J-deficient mice.

**Figure supplement 1—source data 1.** Data for *Figure 2—figure supplement 1*.

*2013*). The results showed that RBP-J deficiency did not affect egress of Ly6C$^{lo}$ monocytes from BM (*Figure 1E*). Taken together, these data revealed RBP-J as a critical regulator controlling homeostasis of peripheral blood Ly6C$^{lo}$ monocytes.

## RBP-J deficiency does not affect Ly6C$^{hi}$ monocyte conversion or Ly6C$^{lo}$ monocyte survival and proliferation

Next, we aimed to determine whether the increase in blood Ly6C$^{lo}$ monocytes in RBP-J-deficient mice was due to decreased cell death or enhanced proliferation. We first stained monocytes with Annexin V and 7-amino-actinomycin D (7-AAD) to identify apoptotic cells and observed comparable percentages of apoptotic blood Ly6C$^{lo}$ monocytes in control and RBP-J-deficient mice (*Figure 2A*). Given the crucial role of Nr4a1 in the survival of Ly6C$^{lo}$ monocytes (*Hanna et al., 2011*), we detected the expression of Nr4a1, which was similar in Ly6C$^{lo}$ monocytes from control and RBP-J-deficient mice (*Figure 2—figure supplement 1A*). We then assessed the proliferative capacity by analyzing Ki-67 expression as well as *in vivo* EdU incorporation. Ki-67 levels in blood monocytes displayed no differences between control and RBP-J-deficient mice (*Figure 2B*). The percentage of EdU$^{+}$ monocytes did not significantly differ between control and RBP-J-deficient mice at the indicated time points, implying that the turnover of Ly6C$^{lo}$ monocytes was normal in RBP-J-deficient mice (*Figure 2C*). Fluorescent latex beads as particulate tracers could be phagocytosed by monocytes after intravenous injection and stably label Ly6C$^{lo}$ monocytes (*Tacke et al., 2006*). Thus, we intravenously injected fluorescent latex beads into control and RBP-J-deficient mice to track circulating monocytes (*Figure 2D*). By day 7 post injection, only Ly6C$^{lo}$ monocytes were latex$^{+}$ as previously reported (*Tacke et al., 2006*), whereas control and RBP-J-deficient mice groups presented a similar frequency of latex$^{+}$ monocytes (*Figure 2E*). Together, these data indicated that RBP-J did not influence monocyte survival and proliferation.

Previous studies have shown that Ly6C$^{lo}$ monocytes are observed in recipient mice following the adoptive transfer of Ly6C$^{hi}$ monocytes, indicating that Ly6C$^{hi}$ monocytes can convert into Ly6C$^{lo}$ monocytes (*Varol et al., 2007*; *Yona et al., 2013*). We next wished to evaluate whether conversion of Ly6C$^{hi}$ monocyte was regulated by RBP-J by isolating BM GFP$^{+}$Ly6C$^{hi}$ monocytes from *Lyz2$^{cre/cre}$Cx3cr1$^{gfp/+}$* control or *Rbpj$^{fl/fl}$Lyz2$^{cre/cre}$Cx3cr1$^{gfp/+}$* mice and adoptively transferring them into *Ccr2$^{RFP/RFP}$* recipients (*Figure 2F*). Sixty hours after transfer, a subset of cells from donors were converted into Ly6C$^{lo}$ monocytes (*Figure 2—figure supplement 1B*), and equal percentages of Ly6C$^{lo}$ monocytes were derived from control and *Rbpj$^{fl/fl}$Lyz2$^{cre/cre}$Cx3cr1$^{gfp/+}$* donors (*Figure 2G*). Thus, the conversion of Ly6C$^{hi}$ monocyte into Ly6C$^{lo}$ monocyte was not affected by RBP-J deficiency.

## RBP-J regulates blood Ly6C$^{lo}$ monocytes in a cell-intrinsic manner

We next wondered whether the increase in Ly6C$^{lo}$ monocytes in RBP-J-deficient mice was BM-derived and cell-intrinsic. We performed BM transplantation by engrafting lethally irradiated mice with a 1:4 mixture of *Rbpj$^{+/+}$Lyz2$^{cre/cre}$* control and *Rbpj$^{fl/fl}$Lyz2$^{cre/cre}$* BM cells (CD45.2) and *Cx3cr1$^{gfp/+}$* BM cells (CD45.1) (*Figure 3A*). Eight weeks after transplantation, we analyzed the frequencies of CD45.2$^{+}$ donor cells in the BM and blood of recipient mice and found that the frequencies of CD45.2$^{+}$ cells within total cells were similar to the mixture ratio of BM cells for Ly6C$^{hi}$ monocytes and neutrophils in both BM and blood (*Figure 3B*, *Figure 3—figure supplement 1A and B*). Specifically, more blood Ly6C$^{lo}$ monocytes were derived from RBP-J-deficient donors than control donors (*Figure 3B*), reflecting that the contribution of RBP-J-deficient cells in blood Ly6C$^{lo}$ cells was significantly higher than that of control cells. These results implied that RBP-J-deficient BM cells were highly efficient in generating blood Ly6C$^{lo}$ monocytes.

Given that RBP-J-deficient mice had more Ly6C$^{lo}$ monocytes in blood than did control mice, we performed parabiosis experiments, a surgical union of two organisms, which allowed parabiotic mice to share their blood circulation. The contribution of circulating cells from one animal to another can be estimated by measuring the percentage of blood cells that originated from each animal (*Liu et al., 2007*). We joined a CD45.1, *Cx3cr1$^{gfp/+}$* mouse with an age- and sex-matched *Rbpj$^{+/+}$Lyz2$^{cre/cre}$* control or *Rbpj$^{fl/fl}$Lyz2$^{cre/cre}$* mouse (CD45.2) (*Figure 3C*). Four weeks after the procedure, about 50% of B cells and T cells in parabiotic mice displayed efficient exchange of their circulation (*Figure 3—figure supplement 1C*). As expected, RBP-J-deficient mice exhibited higher percentages of Ly6C$^{lo}$ monocytes than control animals (*Figure 3D*). Intriguingly, in the parabiotic *Cx3cr1$^{gfp/+}$* mice, RBP-J-deficient cells still constituted significantly higher proportion of circulating Ly6C$^{lo}$, but not Ly6C$^{hi}$ monocytes,

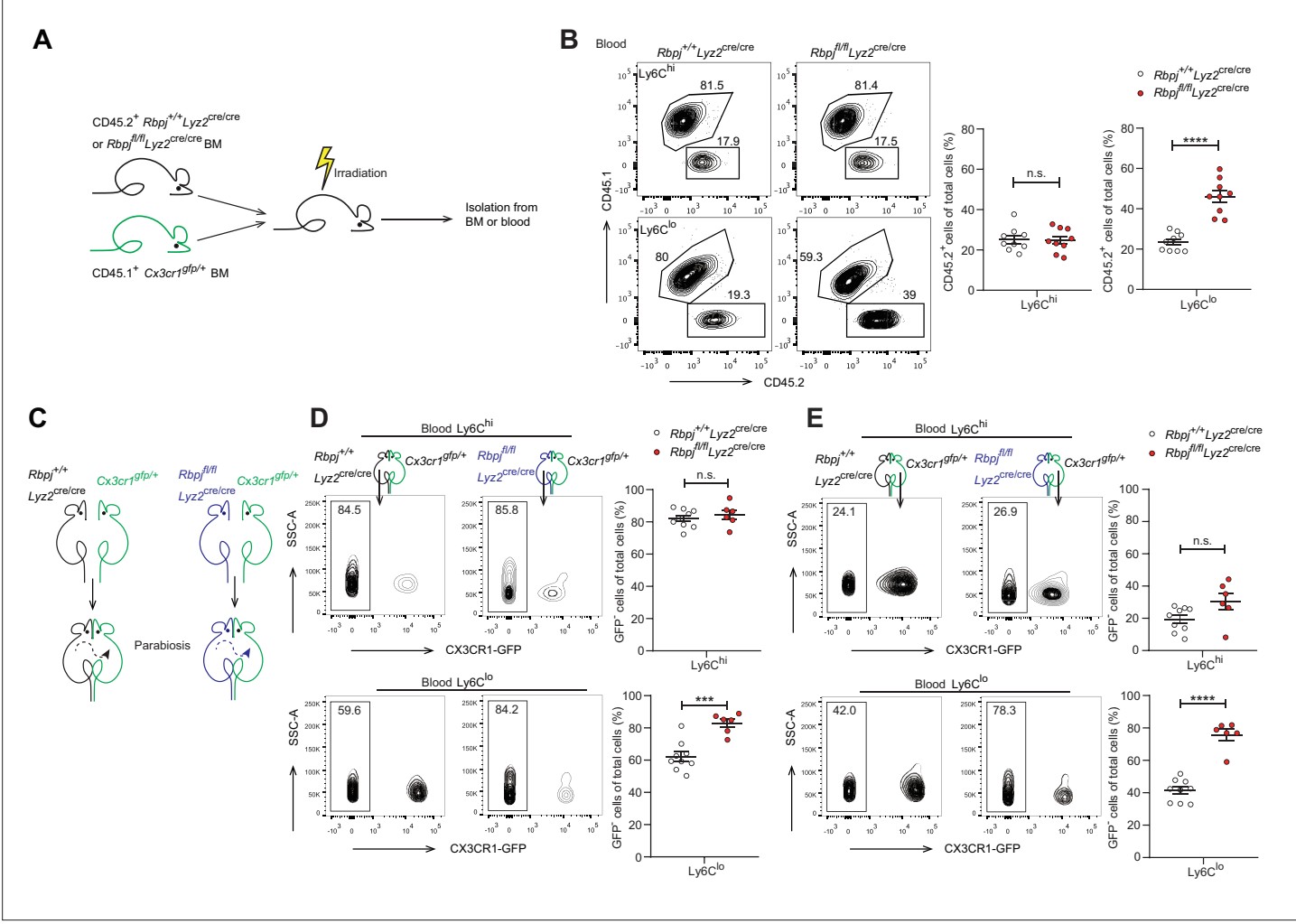

**Figure 3.** The role of RBP-J in blood Ly6C$^{lo}$ monocytes is cell-intrinsic. (**A**) Cartoon depicting the bone marrow (BM) transplantation. BM recipient 6-week-old male C57BL/6 mice (CD45.2) were lethally irradiated. BM cells from donor mice (*Rbpj$^{+/+}$Lyz2$^{cre/cre}$* or *Rbpj$^{fl/fl}$Lyz2$^{cre/cre}$* and CD45.1, *Cx3cr1$^{gfp/+}$*) were collected and transferred into recipient mice. Mice were used after 8 wk of BM reconstitution. (**B**) Representative FACS plots (left) and cumulative data (right) quantitating the frequency of *Rbpj$^{+/+}$Lyz2$^{cre/cre}$* and *Rbpj$^{fl/fl}$Lyz2$^{cre/cre}$* donor cells among Ly6C$^{hi}$ and Ly6C$^{lo}$ monocytes in the blood of recipient mice. (**C**) Cartoon depicting the generation of *Rbpj$^{+/+}$Lyz2$^{cre/cre}$* control or *Rbpj$^{fl/fl}$Lyz2$^{cre/cre}$* and *Cx3cr1$^{gfp/+}$* parabiotic pairs. (**D**) Representative FACS plots (left) and cumulative data (right) quantitating percentages of monocyte subsets derived from control or RBP-J-deficient mice in control or RBP-J-deficient mice. (**E**) Representative FACS plots (left) and cumulative data (right) quantitating percentages of monocyte subsets derived from control or RBP-J-deficient mice in *Cx3cr1$^{gfp/+}$* mice. Data are pooled from at least two independent experiments; n ≥ 6 in each group. Data are shown as mean ± SEM; n.s., not significant; ***p<0.001; ****p<0.0001 (two-tailed Student's unpaired *t*-test). Each symbol represents an individual mouse. SSC-A, side scatter area.

The online version of this article includes the following source data and figure supplement(s) for figure 3:

**Source data 1.** Data of *Figure 3*.

**Figure supplement 1.** Cell-intrinsic requirement of RBP-J for Ly6C$^{lo}$ monocytes maintenance.

**Figure supplement 1—source data 1.** Data for *Figure 3—figure supplement 1*.

than RBP-J sufficient counterparts as indicated by the percentages of the GFP-negative population (*Figure 3E*). These results implicated that the enhanced ability of Ly6C$^{lo}$ monocytes to circulate in the peripheral blood as a result of RBP-J deficiency was cell-intrinsic.

## RBP-J regulates phenotypical marker genes in blood Ly6C$^{lo}$ monocytes

To further study the consequences of RBP-J loss of function in blood Ly6C$^{lo}$ monocytes, we performed gene expression profiling by RNA-seq, which revealed that Ly6C$^{lo}$ monocytes in RBP-J-deficient mice

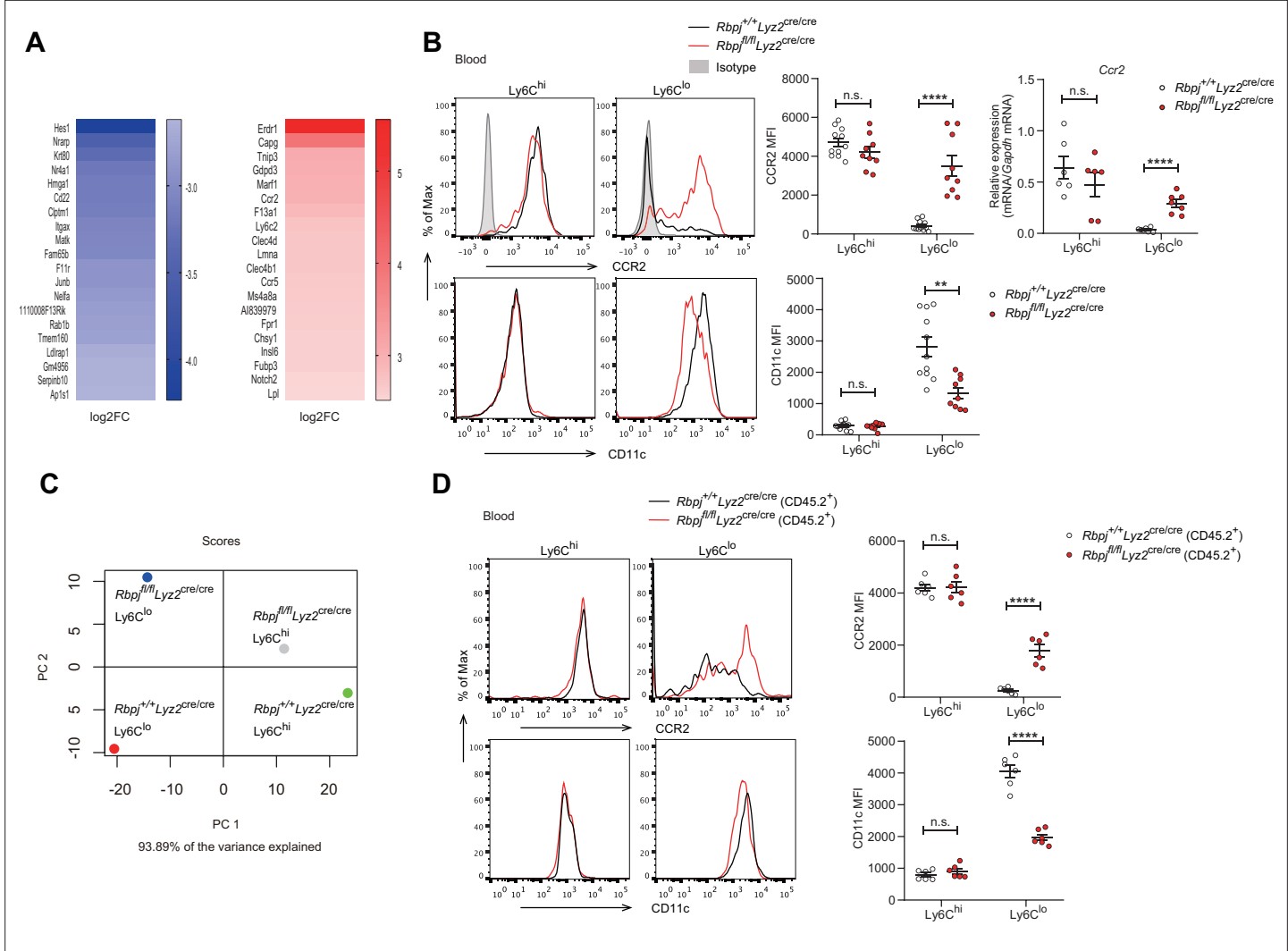

**Figure 4.** Phenotypic markers in Ly6C$^{lo}$ monocytes are changed in RBP-J-deficient mice. (**A**) Heatmap of RNA-seq dataset showing the top 20 downregulated and upregulated genes in blood Ly6C$^{lo}$ monocytes from *Rbpj*$^{fl/fl}$*Lyz2*$^{cre/cre}$ versus *Rbpj*$^{+/+}$*Lyz2*$^{cre/cre}$ control mice. Blue and red font indicates downregulated and upregulated genes in RBP-J-deficient Ly6C$^{lo}$ monocytes respectively. (**B**) Representative FACS plots, cumulative mean fluorescence intensity (MFI), and quantitative real-time PCR (qPCR) analysis of CCR2/CD11c expression in control and RBP-J-deficient blood monocyte subsets. Shaded curves represent isotype control, black lines represent control mice, and red lines represent RBP-J-deficient mice. (**C**) Principal component analysis (PCA) of indicated cell types. (**D**) Representative FACS plots and cumulative MFI of CCR2/CD11c expression in blood monocyte subsets derived from control or RBP-J-deficient mice. Black lines represent control mice, and red lines represent RBP-J-deficient mice. Data are pooled from two independent experiments (**B, D**); n ≥ 6 in each group. Data are shown as mean ± SEM; n.s., not significant; **p<0.01; ****p<0.0001 (two-tailed Student's unpaired *t*-test). Each symbol represents an individual mouse.

The online version of this article includes the following source data and figure supplement(s) for figure 4:

**Source data 1.** Data of *Figure 4*.

**Figure supplement 1.** Normal expression of CCR2 and CD11c in bone marrow (BM) monocytes.

**Figure supplement 1—source data 1.** Data for *Figure 4—figure supplement 1*.

exhibited low expression of *Itgax* but high expression of *Ccr2*, in comparison to those in control mice (***Figure 4A***). These differential expression patterns of phenotypic marker genes were also confirmed by qPCR and flow cytometry analysis (***Figure 4B***). However, blood Ly6C$^{hi}$ monocytes and BM monocytes displayed normal levels of CD11c and CCR2 (***Figure 4B***, ***Figure 4—figure supplement 1A***). Moreover, principal component analysis showed distinct expression pattern of monocytes between control and RBP-J-deficient mice (***Figure 4C***). To determine whether the regulation of phenotypic markers by RBP-J in blood Ly6C$^{lo}$ monocytes was cell-intrinsic, we performed BM transplantation

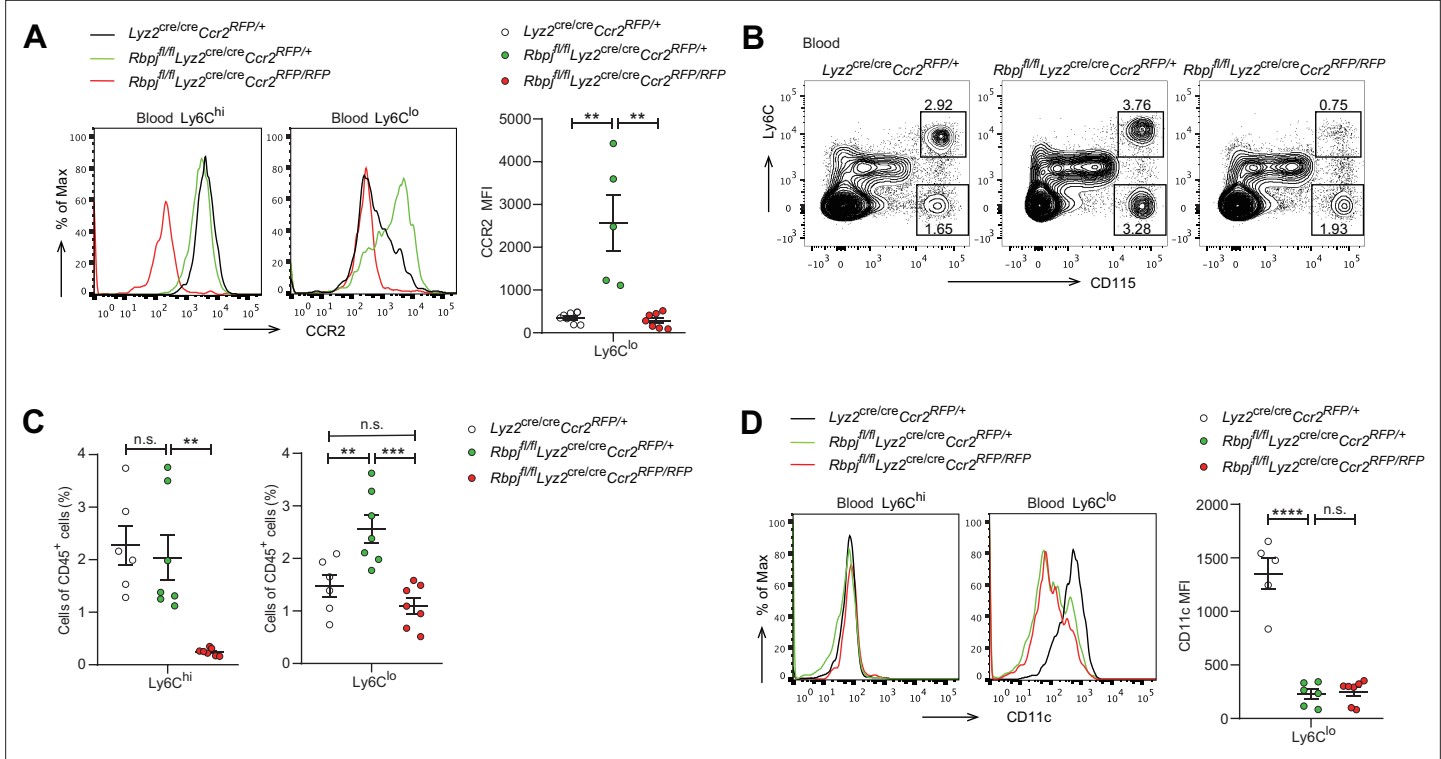

**Figure 5.** Blood Ly6C[lo] monocytes are decreased in double-deficient (DKO) mice. (**A**) Representative FACS plots and cumulative mean fluorescence intensity (MFI) of CCR2 expression in *Lyz2*[cre/cre]*Ccr2*[RFP/+] control, *Rbpj*[fl/fl]*Lyz2*[cre/cre]*Ccr2*[RFP/+] and *Rbpj*[fl/fl]*Lyz2*[cre/cre]*Ccr2*[RFP/RFP] (DKO) blood Ly6C[hi] and Ly6C[lo] monocytes are shown. (**B, C**) Blood monocyte subsets in control, RBP-J-deficient, and DKO mice were determined by FACS. Representative FACS plots (**B**) and cumulative data of cell ratio (**C**) are shown. (**D**) Representative FACS plots and cumulative MFI of CD11c expression in control, RBP-J-deficient and DKO blood Ly6C[hi] and Ly6C[lo] monocytes are shown. Data are pooled from at least two independent experiments; n ≥ 5 in each group. Data are shown as mean ± SEM; n.s., not significant; **p<0.01; ***p<0.001; ****p<0.0001 (two-tailed Student's unpaired *t*-test). Each symbol represents an individual mouse.

The online version of this article includes the following source data for figure 5:

**Source data 1.** Data for *Figure 5*.

as described above (*Figure 3A*). The findings indicated that Ly6C[lo] monocytes derived from RBP-J-deficient BM cells expressed high levels of CCR2 but low levels of CD11c, in comparison to those derived from control BM cells, whereas no significant changes were observed in blood Ly6C[hi] and BM monocytes, as expected (*Figure 4D*, *Figure 4—figure supplement 1B*). These results collectively suggested that RBP-J regulated the expression of CCR2/CD11c in a cell-intrinsic manner.

### RBP-J-mediated control of blood Ly6C[lo] monocytes is CCR2 dependent

Given that RBP-J deficiency led to enhanced CCR2 expression in Ly6C[lo] monocytes and that CCR2 is essential for monocyte functionality under various inflammatory and non-inflammatory conditions (*Shi and Pamer, 2011*), we wished to genetically investigate the role of CCR2 in the RBP-J-deficient background. We generated RBP-J/CCR2 double-deficient (DKO) mice with the genotype *Rbpj*[fl/fl]*Lyz2*[cre/cre]*Ccr2*[RFP/RFP] with *Lyz2*[cre/cre]*Ccr2*[RFP/+] and *Rbpj*[fl/fl]*Lyz2*[cre/cre]*Ccr2*[RFP/+] mice serving as controls. As expected, the expression of CCR2 in Ly6C[lo] monocytes was reduced in DKO mice compared to RBP-J-deficient mice (*Figure 5A*), confirming the successful deletion of the *Ccr2* gene. DKO mice showed a lower percentage of both Ly6C[hi] and Ly6C[lo] monocytes than RBP-J-deficient mice (*Figure 5B and C*). Notably, the percentage of Ly6C[lo] monocytes in DKO mice was comparable to that observed in the control mice (*Figure 5B and C*), implicating that deletion of CCR2 corrected RBP-J deficiency-associated phenotype of increased Ly6C[lo] monocytes. Whereas, both RBP-J-deficient and DKO mice exhibited lower expression levels of CD11c in their Ly6C[lo] monocyte than control mice, suggesting that the regulation of CD11c expression was independent of CCR2 (*Figure 5D*). These results suggested that RBP-J regulated Ly6C[lo] monocytes, at least in part, through CCR2.

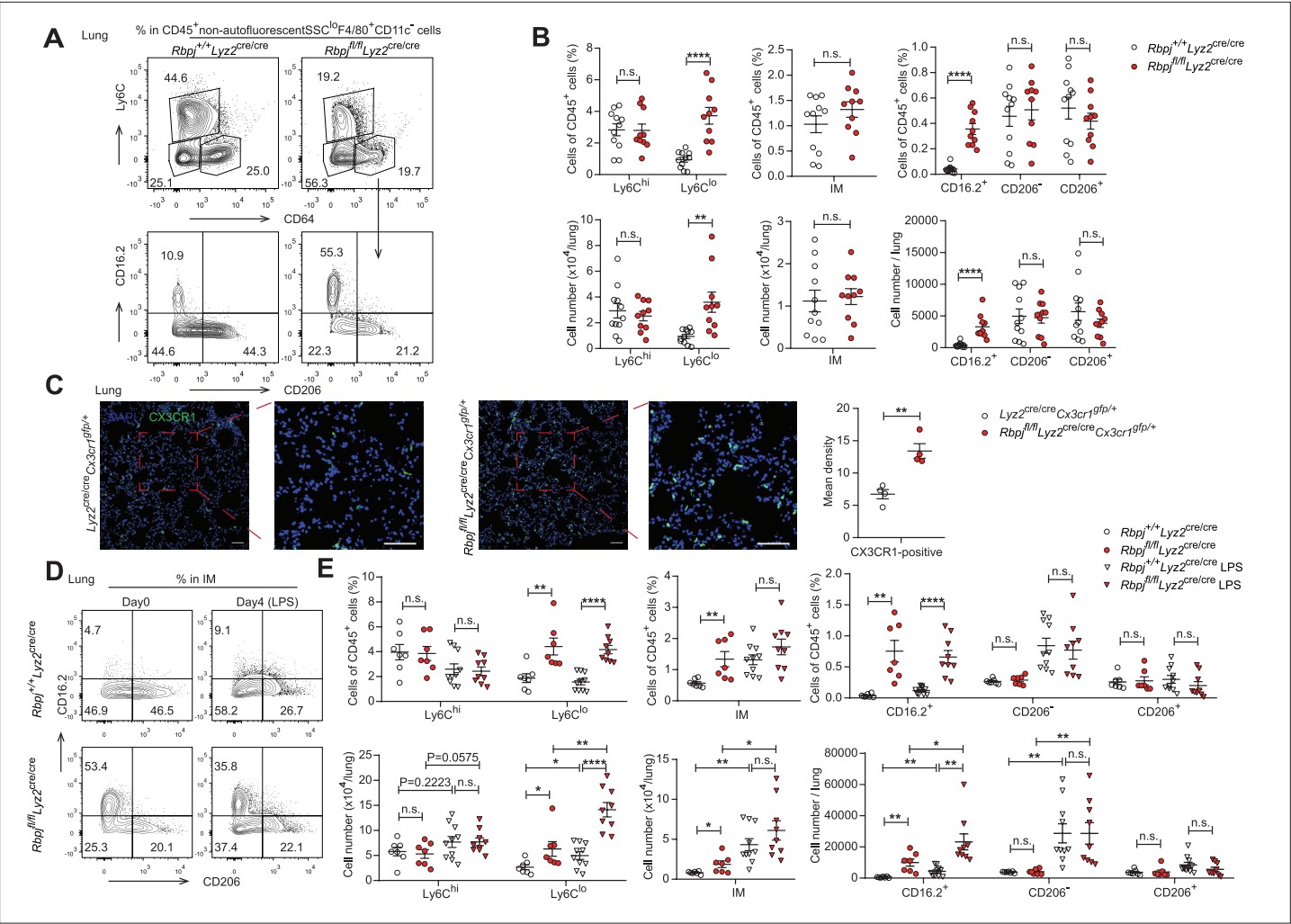

**Figure 6.** RBP-J-deficient mice exhibit more lung Ly6C$^{lo}$ monocytes and CD16.2$^+$ interstitial macrophages (IM). (**A, B**) indicate populations in the lungs of *Rbpj$^{+/+}$Lyz2$^{cre/cre}$* and *Rbpj$^{fl/fl}$Lyz2$^{cre/cre}$* mice were determined by FACS. Representative FACS plots (**A**) and cumulative data of cell ratio and absolute numbers (**B**) are shown. (**C**) Immunofluorescence staining for GFP$^+$ cells in the lungs from *Lyz2$^{cre/cre}$Cx3cr1$^{gfp/+}$* and *Rbpj$^{fl/fl}$Lyz2$^{cre/cre}$Cx3cr1$^{gfp/+}$* mice (CX3CR1 [green]; DAPI [blue]). Scale bars represent 50 μm. (**D, E**) *Rbpj$^{+/+}$Lyz2$^{cre/cre}$* and *Rbpj$^{fl/fl}$Lyz2$^{cre/cre}$* mice were instilled intranasally with phosphate buffered saline (PBS) or PBS containing lipopolysaccharide (LPS), and lungs were harvested at the indicated time points. Representative FACS plots (**D**) and cumulative data of cell ratio and absolute numbers (**E**) are shown. Data are pooled from at least two independent experiments; n ≥ 4 in each group. Data are shown as mean ± SEM; n.s., not significant; *p<0.05; **p<0.01; ****p<0.0001 (two-tailed Student's unpaired *t*-test). Each symbol represents an individual mouse.

The online version of this article includes the following source data and figure supplement(s) for figure 6:

**Source data 1.** Data for *Figure 6*.

**Source data 2.** Data for *Figure 6C*.

**Figure supplement 1.** RBP-J is not required for turnover of lung Ly6C$^{lo}$ monocytes and CD16.2$^+$ interstitial macrophages (IM).

**Figure supplement 1—source data 1.** Data for *Figure 6—figure supplement 1*.

## Lung Ly6C$^{lo}$ monocytes are accumulated in RBP-J-deficient mice

In mice, alveolar macrophages (AM) are maintained by local self-renewal and arise from fetal liver-derived precursors, whereas interstitial macrophages (IM) probably originate from monocytes (*Guilliams et al., 2013*; *Hashimoto et al., 2013*; *Sabatel et al., 2017*). Schyns et al. identified three subpopulations of IM according to expression of CD206 and CD16.2, and Ly6C$^{lo}$ monocytes are proposed to give rise to CD64$^+$CD16.2$^+$ IM (*Schyns et al., 2019*). We next compared the populations of lung monocytes and IM between control and RBP-J-deficient mice. The conditional deletion of RBP-J resulted in a significant increase in the absolute and relative numbers of Ly6C$^{lo}$ monocytes and

CD16.2+ IM, while the numbers of Ly6Chi monocytes, CD206- IM, and CD206+ IM remained unchanged (*Figure 6A and B*). To confirm these findings, lung sections from *Lyz2*cre/cre*Cx3cr1*gfp/+ control and *Rbpj*fl/fl*Lyz2*cre/cre*Cx3cr1*gfp/+ mice were stained with an anti-GFP antibody, which revealed a significant increase in the number of GFP+ cells in RBP-J-deficient mice (*Figure 6C*). In addition, we evaluated the proliferative capacity of Ly6Clo monocytes and CD16.2+ IM, but did not observe enhanced EdU incorporation in Ly6Clo monocytes and CD16.2+ IM from RBP-J-deficient mice (*Figure 6—figure supplement 1A and B*). These findings suggested that the augmented population of these cells might not have resulted from increased *in situ* proliferation. Previous reports have indicated that lipopolysaccharide (LPS) can increase the population of IM in a CCR2-dependent manner (*Sabatel et al., 2017*). We thus challenged mice with LPS and analyzed monocytes and IM at days 0 and 4. LPS exposure induced an increase in numbers of Ly6Clo monocytes, CD16.2+ IM, and CD206- IM compared to baseline both in control and RBP-J-deficient mice (*Figure 6D and E*). There was a trend toward higher Ly6Chi monocyte after LPS treatment, although this observation was not statistically significant. Of note, RBP-J-deficient

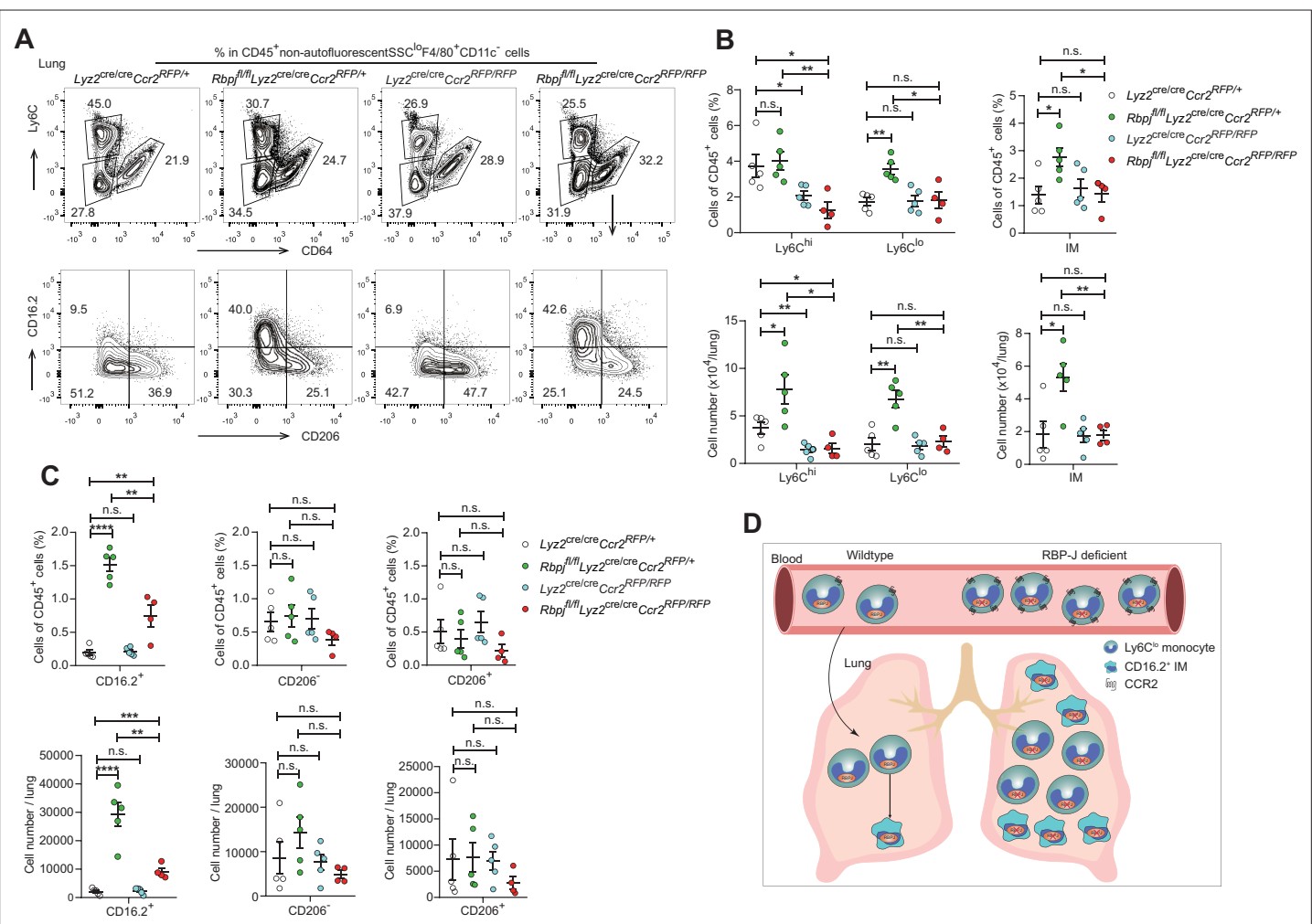

**Figure 7.** Double-deficient (DKO) mice lack lung Ly6Clo monocytes and CD16.2+ interstitial macrophages (IM). (**A**) Representative FACS plots of lung monocyte and IM subsets in *Lyz2*cre/cre*Ccr2*RFP/+, *Rbpj*fl/fl*Lyz2*cre/cre*Ccr2*RFP/+, *Lyz2*cre/cre*Ccr2*RFP/RFP and *Rbpj*fl/fl*Lyz2*cre/cre*Ccr2*RFP/RFP mice. (**B, C**) Cumulative data of cell ratio and absolute numbers of monocyte (**B**) and IM (**C**) subsets. (**D**) Proposed model. RBP-J is a crucial regulator of blood Ly6Clo monocytes. Mice with conditional deletion of RBP-J in myeloid cells exhibit a marked increase in blood Ly6Clo monocytes, which highly express CCR2, and subsequently accumulate lung Ly6Clo monocytes and CD16.2+ IM. Data are pooled from two independent experiments; n ≥ 4 in each group. Data are shown as mean ± SEM; n.s., not significant; *p<0.05; **p<0.01; ***p<0.001; ****p<0.0001 (two-tailed Student's unpaired *t*-test). Each symbol represents an individual mouse.

The online version of this article includes the following source data for figure 7:

**Source data 1.** Data for *Figure 7*.

mice exhibited robustly elevated numbers of Ly6C$^{lo}$ monocytes and CD16.2$^+$ IM compared with control mice after LPS treatment (*Figure 6D and E*), suggesting that RBP-J-deficient Ly6C$^{lo}$ monocytes were recruited to inflamed tissue in large numbers and differentiated into CD16.2$^+$ IM.

To investigate whether the elevated levels of lung monocytes and CD16.2$^+$ IM in RBP-J-deficient mice were derived from blood Ly6C$^{lo}$ monocytes, we examined monocytes and IM in *Lyz2*$^{cre/cre}$*Ccr2*$^{RFP/+}$ control, *Lyz2*$^{cre/cre}$*Ccr2*$^{RFP/RFP}$, *Rbpj*$^{fl/fl}$*Lyz2*$^{cre/cre}$*Ccr2*$^{RFP/+}$ and *Rbpj*$^{fl/fl}$*Lyz2*$^{cre/cre}$*Ccr2*$^{RFP/RFP}$ (DKO) mice. The deletion of CCR2 led to a decrease in Ly6C$^{hi}$ monocytes, and the Ly6C$^{lo}$ monocytes in DKO mice were reduced to a level similar to that of control mice (*Figure 7A and B*). While the DKO mice had more CD16.2$^+$ IM compared to control mice, these cells were severely reduced compared to RBP-J-deficient mice (*Figure 7A and C*). The above data supported the notion that increased lung Ly6C$^{lo}$ monocytes and CD16.2$^+$ IM in RBP-J-deficient mice were derived from blood Ly6C$^{lo}$ monocytes. In summary, these results suggested that in RBP-J-deficient mice, recruitment of blood Ly6C$^{lo}$ monocytes to the lung was markedly facilitated by increased cell number as well as heightened expression of CCR2, leading to the increase of lung Ly6C$^{lo}$ monocytes and CD16.2$^+$ IM.

## Discussion

In this study, we identified RBP-J as a pivotal factor in regulating blood Ly6C$^{lo}$ monocyte cell fate. Our results showed that mice with conditional deletion of RBP-J in myeloid cells exhibited a robust increase in blood Ly6C$^{lo}$ monocytes and subsequently accumulated lung Ly6C$^{lo}$ monocytes and CD16.2$^+$ IM under steady-state conditions (*Figure 7D*). BM transplantation experiments in which RBP-J-deficient cells were transplanted into recipient mice as well as the parabiosis experiment showed a similar increase in blood Ly6C$^{lo}$ monocytes, demonstrating that RBP-J was an intrinsic factor required for the regulation of Ly6C$^{lo}$ monocytes. Further analysis revealed that RBP-J regulated the expression of CCR2 and CD11c on the surface of Ly6C$^{lo}$ monocytes. Moreover, the phenotype of elevated blood Ly6C$^{lo}$ monocytes and their progeny was further ameliorated by deleting *Ccr2* in an RBP-J-deficient background.

Notch-RBP-J signaling has been shown to play a role in regulating functional polarization and activation of macrophages, and regulated formation of Kupffer cells and macrophage differentiation from Ly6C$^{hi}$ monocytes in ischemia (*Foldi et al., 2016*; *Hu et al., 2008*; *Kang et al., 2020*; *Krishnasamy et al., 2017*; *Sakai et al., 2019*; *Wang et al., 2010*; *Xu et al., 2012*). At steady state, interaction of DLL1 with Notch 2 regulates conversion of Ly6C$^{hi}$ monocytes into Ly6C$^{lo}$ monocytes in special niches of the BM and spleen (*Gamrekelashvili et al., 2016*). Under inflammatory conditions, Notch2 and TLR7 pathways independently and synergistically promote conversion of Ly6C$^{hi}$ monocytes into Ly6C$^{lo}$ monocytes (*Gamrekelashvili et al., 2020*). However, in our study, the percentage of blood Ly6C$^{lo}$ monocytes increased in RBP-J-deficient mice. These results somewhat differ from what was observed in mice with conditional deletion of Notch2 (*Gamrekelashvili et al., 2016*). Actually, four members of the Notch family have been identified in mammals (*Radtke et al., 2013*). Thus, Notch receptors seem to be non-redundant in regulating monocyte cell fate and distinct Notch receptors are likely to act in a concerted manner to coordinate the monocyte differentiation program.

Colony-stimulating factor 1 receptor (CSF1R) signal, CX3CR1, CEBPβ, and Nr4a1 have been suggested to be involved in the generation and survival of Ly6C$^{lo}$ monocytes. The lifespan of Ly6C$^{lo}$ monocytes is acutely shortened after blockade of CSF1R, and the numbers of these cells are reduced, and the percentage of dead cells increased in mice with endothelial cell-specific depletion of *Csf1* (*Emoto et al., 2022*; *MacDonald et al., 2010*). CX3CR1 and CEBPβ-knockout mice show accelerated death of Ly6C$^{lo}$ monocyte and display the decreased level of Ly6C$^{lo}$ monocytes (*Landsman et al., 2009*; *Tamura et al., 2017*). Nr4a1 knockout mice have significantly fewer monocytes in blood, BM, and spleen due to differentiation deficiency in MDP and accelerated apoptosis of Ly6C$^{lo}$ monocytes in BM (*Hanna et al., 2011*). RBP-J-deficient mice had normal MDP and Ly6C$^{lo}$ monocytes, expressed normal level of Nr4a1 and Ki-67, and exhibited normal half-life and turnover rate, which suggested that RBP-J may not regulate Ly6C$^{lo}$ monocytes through these factors. Further studies are required to elucidate the precise molecular mechanisms of RBP-J in Ly6C$^{lo}$ monocytes under steady state.

At the steady state, Ly6C$^{lo}$ monocytes patrol endothelium of blood vessel. After R848-induced endothelial injury, Ly6C$^{lo}$ monocytes recruit neutrophils to mediate focal endothelial necrosis and fuel vascular inflammation, and subsequently, the Ly6C$^{lo}$ monocytes remove cellular debris (*Carlin et al., 2013*; *Imhof et al., 2016*). In response to *Listeria monocytogenes* infection, Ly6C$^{lo}$ monocytes can

extravasate but do not give rise to macrophages or DCs (*Auffray et al., 2007*). Additional evidence shows that Ly6C$^{lo}$ monocytes do not merely act as luminal blood macrophages. In lung, exposure to unmethylated CpG DNA expands CD16.2$^+$ IM, which originate from Ly6C$^{lo}$ monocytes and spontaneously produce IL-10, thereby preventing allergic inflammation (*Schyns et al., 2019*). A recent study shows that Ly6C$^{lo}$ monocytes give rise to CD9$^+$ macrophages, which provide an intracellular replication niche for *Salmonella* Typhimurium in the spleen, and Ly6C$^{lo}$-depleted mice are more resistant to *Salmonella* Typhimurium infection, suggesting that Ly6C$^{lo}$ monocytes exert certain functions in systemic infection (*Hoffman et al., 2021*). Our data provide evidence for RBP-J in the function of blood Ly6C$^{lo}$ monocytes as RBP-J-deficient Ly6C$^{lo}$ monocytes exhibited enhanced competition in blood circulation, as well as gave rise to increased numbers of lung CD16.2$^+$ IM. In summary, we showed that RBP-J acted as a fundamental regulator of the maintenance of blood Ly6C$^{lo}$ monocytes and their descendent, at least in part through regulation of CCR2. These results provide insights into understanding the mechanisms that regulate monocyte homeostasis and function.

# Materials and methods

## Mice

*Cx3cr1$^{gfp/gfp}$* mice (JAX stock 005582) and *Ccr2$^{RFP/RFP}$* mice (JAX stock 017586) were purchased from the Jackson Laboratory. Mice with a myeloid-specific deletion of the *Rbpj* were generated by crossing *Rbpj$^{fl/fl}$* mice to *Lyz2*-Cre mice as described previously (*Hu et al., 2008*). *Rbpj$^{fl/fl}$Lyz2$^{cre/cre}$* were crossed to *Cx3cr1$^{gfp/gfp}$* mice to obtain *Lyz2$^{cre/cre}$Cx3cr1$^{gfp/+}$* and *Rbpj$^{fl/fl}$Lyz2$^{cre/cre}$Cx3cr1$^{gfp/+}$* mice. Cx3cr1$^{gfp/+}$ mice were obtained by crossing *Cx3cr1$^{gfp/gfp}$* with C57/BL6 CD45.1$^+$ mice. *Rbpj$^{fl/fl}$Lyz2$^{cre/cre}$* mice were crossed with *Ccr2$^{RFP/RFP}$* mice to obtain *Lyz2$^{cre/cre}$Ccr2$^{RFP/+}$*, *Lyz2$^{cre/cre}$Ccr2$^{RFP/RFP}$*, *Rbpj$^{fl/fl}$Lyz2$^{cre/cre}$Ccr2$^{RFP/+}$* and *Rbpj$^{fl/fl}$Lyz2$^{cre/cre}$Ccr2$^{RFP/RFP}$* mice. All mice were maintained under specific pathogen-free conditions. All animal experimental protocols were approved by the Institutional Animal Care and Use Committees of Tsinghua University (17-hxy). Gender- and age-matched mice were used at 7–12 weeks old for experiments.

## Quantitative RT-PCR

Blood monocytes were sorted by flow cytometry. The total RNA was extracted using total RNA purification kit (GeneMarkbio) and reversely transcribed to cDNA by M-MLV Reverse Transcriptase (Takara). qPCR was performed on a real-time PCR system (StepOnePlus; Applied Biosystems) using FastSYBR mixture (CWBIO). *Gapdh* messenger RNA was used as internal control to normalize the expression of target genes. Primer sequences are provided in *Table 1*.

## RNA-seq analysis

Ly6C$^{hi}$ and Ly6C$^{lo}$ blood monocytes were isolated from *Rbpj$^{+/+}$Lyz2$^{cre/cre}$* and *Rbpj$^{fl/fl}$Lyz2$^{cre/cre}$* mice, and total RNA was extracted using total RNA purification kit (GeneMarkbio). RNA was converted into RNA-seq libraries, which were sequenced with the pair-end option using an Illumina-HiSeq2500 platform at Beijing Genomics Institute (BGI), China. The significantly downregulated genes were identified with p-value<0.05 and (FPKM + 1) fold changes ≤ 0.2, and significantly upregulated genes were identified with p-value <0.05 and (FPKM + 1) fold changes ≥ 5.7. The RNA-seq data are deposited in Gene Expression Omnibus under accession number GSE208772.

**Table 1.** Primers sequences for regular quantitative real-time PCR (qPCR) used in this study.

| Gene | Forward primer | Reverse primer |
| --- | --- | --- |
| *Gapdh* | ATCAAGAAGGTGGTGAAGCA | AGACAACCTGGTCCTCAGTGT |
| *Rbpj* | ACCCCTGTGCCTGTCGTAGAA | TCCCGGAATGCAGAAATGTC |
| *Nr4a1* | TTGAGCTTGAATACAGGGCA | AGTTGGGGGAGTGTGCTAGA |
| *Ccr2* | CCTTGGGAATGAGTAACTGTGTGAT | ATGGAGAGATACCTTCGGAACTTCT |

## Annexin V staining

Annexin V and 7-AAD were used for identification of apoptotic monocytes by flow cytometry. Blood cells were stained with annexin V (eBioscience) and 7-AAD (BioLegend) according to the manufacturer's protocols.

## EdU pulsing and latex beads labeling

Mice were injected intravenously with a single 1 mg EdU (Thermo Scientific). BM and blood cells were collected and stained with fluorescence-conjugated mAb against CD45, CD11b, Ly6G, CD115, and Ly6C. Cells were then fixed, permeabilized, and stained with reaction cocktail using EdU Assay Kit. Labeled cells were analyzed by flow cytometry.

Mice were injected intravenously with a single 10 µl latex beads (Polysciences) in 250 µl phosphate buffered saline (PBS). Blood cells were harvested at indicated time. Cells were then stained for CD45, CD11b, Ly6G, CD115, Ly6C, and analyzed by flow cytometry.

## *In vivo* labeling of sinusoidal leukocytes

Mice were injected intravenously with 1 µg of PE-conjugated anti-CD45 antibodies. Two minutes after antibody injection, mice were sacrificed, and BM were harvested. BM cells were then stained and analyzed by flow cytometry.

## Generation of BM chimera

C57/BL6 CD45.2$^+$ mice were lethally irradiated in two doses of 5.5 Gy 2 hr apart. $0.8 \times 10^5$ BM cells from $Rbpj^{+/+}Lyz2^{cre/cre}$ mice (CD45.2$^+$) or $Rbpj^{fl/fl}Lyz2^{cre/cre}$ mice (CD45.2$^+$) were mixed with $3.2 \times 10^5$ BM cells from $Cx3cr1^{gfp/+}$ mice (CD45.1$^+$), and injected intravenously into recipient mice. Mice were used for experiments 8 wk after irradiation.

## Adoptive transfers

BM GFP$^+$Ly6C$^{hi}$ monocytes were sorted from $Lyz2^{cre/cre}Cx3cr1^{gfp/+}$ or $Rbpj^{fl/fl}Lyz2^{cre/cre}Cx3cr1^{gfp/+}$ mice, and transferred to $Ccr2^{RFP/RFP}$ mice. Blood cells were collected 60 hr later for flow cytometry analyses.

## Parabiosis

$Cx3cr1^{gfp/+}$ CD45.1$^+$ mice were surgically joined with age-matched female $Rbpj^{fl/fl}Lyz2^{cre/cre}$ or $Rbpj^{+/+}$-$Lyz2^{cre/cre}$ CD45.2$^+$ mice at the age of 6 wk. Bloods were obtained via cardiac puncture, and cell populations were analyzed by flow cytometry at 4 wk after the surgery.

## Intranasal instillations of LPS

$Rbpj^{+/+}Lyz2^{cre/cre}$ or $Rbpj^{fl/fl}Lyz2^{cre/cre}$ mice were anesthetized with isoflurane and intranasally instilled with 10 µg LPS in 25 µl of PBS. Lungs were harvested 4 d later for flow cytometry analyses.

## Immunofluorescence histology

Lungs from $Rbpj^{fl/fl}Lyz2^{cre/cre}Cx3cr1^{gfp/+}$ and $Lyz2^{cre/cre}Cx3cr1^{gfp/+}$ mice were fixed in 1% paraformaldehyde and incubated in 30% sucrose separately overnight at 4°C. The samples were then incubated in the mixture of 30% sucrose and OCT compound (Sakura Finetek) overnight at 4°C. The tissues were embedded and frozen in OCT compound and then cut at 10 µm thickness. Tissue sections were dried for 10 min at 50°C and then fixed in 1% paraformaldehyde at room temperature for 10 min, permeabilized in PBS/0.5% Triton X-100/0.3 M glycine for 10 min, and blocked in PBS/5% goat serum for 1 hr at room temperature. Sections were stained with rabbit anti-GFP antibodies (1:200; Proteintech) overnight at 4°C and washed with PBS/0.1% Tween-20 for 30 min three times at room temperature. Sections were then incubated with AF488-conjugated goat anti-rabbit antibodies (1:1000; Cell Signaling Technology) for 2 hr at room temperature and washed with PBS/0.1% Tween-20 for 30 min three times at room temperature. Sections were stained with DAPI (Solarbio) for 7 min, washed in PBS for 8 min two times at room temperature, and mounted with SlowFade Diamond Antifade Mountant (Life Technologies).

## Cell isolation and flow cytometry

Peripheral blood (PB) was sampled by eyeball extirpating, spleens were mashed through a 70 μm strainer, and BM cells were collected from femurs. Lungs were perfused with 5 ml of HBSS (MACGENE) through the right ventricle, excised, and digested in HBSS containing 5% FBS, 1 mg/ml collagenase type I (Sigma-Aldrich) and 0.05 mg/ml DNase I (Sigma-Aldrich) for 1 hr at 37°C. The digested tissues were homogenized by shaking, passed through a 70 μm cell strainer to create a single-cell suspension. The suspension was enriched in mononuclear cells and harvested from the 1.080:1.038 g/ml interface using a density gradient (Percoll from GE Healthcare). BM, spleen, PB, and lung cells were stained with fluorescently conjugated antibodies. The absolute number of cells was counted by using CountBright Absolute Counting Beads (Invitrogen).

Antibodies against CD45 (30-F11), Ly6C (HK1.4), CD64 (X54-5/7.1), CD206 (C068C2), CD16.2 (9E9), CD45.1 (A20), F4/80 (BM8), Biotin, CD19 (6D5), CCR2 (SA203G11), and CD3ε (145-2C11) were purchased from BioLegend. Antibodies against Ly6G (1A8-Ly6g), CD11b (M1/70), CD115 (AFS98), CD117 (2B8), CD135 (A2F10), CD11c (N418), CD206 (MR6F3), CD45.2 (104), Nur77 (12.14), Ki-67 (SolA15), CD4 (RM4-5), and lineage maker were purchased from eBioscience. Fluorescence-conjugated mAb against Ly6C (AL-21) were purchased from BD Biosciences. Isotype-matched antibodies (eBioscience) were used for control staining. All antibodies were used in 1:400 dilutions, and surface antigens were stained on ice for 30 min.

For intracellular staining, cells were stained with antibodies to surface antigens, fixed and permeabilized with the Cytofix/Cytoperm Fixation/Permeabilization Solution Kit (BD Biosciences), and stained with antibodies or isotype control diluted in permeabilization buffer separately for 30 min at room temperature.

Cells were analyzed on FACSFortessa or FACSAria III flow cytometer (BD Biosciences) using FlowJo software.

## Statistical analysis

Statistical analysis was performed using GraphPad Prism software. All results are shown as mean ± SEM. Statistical significance was determined using Student's unpaired $t$-test. Statistical significance was defined as $p < 0.05$.

## Acknowledgements

We thank the core facility at Institute for Immunology, Tsinghua University, for valuable technical assistance.

# Additional information

### Competing interests

Xiaoyu Hu: Reviewing editor, eLife. The other authors declare that no competing interests exist.

### Funding

| Funder | Grant reference number | Author |
| --- | --- | --- |
| National Natural Science Foundation of China | 31821003 | Xiaoyu Hu |
| National Natural Science Foundation of China | 31991174 | Xiaoyu Hu |
| National Natural Science Foundation of China | 32030037 | Xiaoyu Hu |
| National Natural Science Foundation of China | 82150105 | Xiaoyu Hu |
| Ministry of Science and Technology of the People's Republic of China | 2020YFA0509100 | Xiaoyu Hu |

| Funder | Grant reference number | Author |
|--------|------------------------|--------|
| Center for Life Sciences | | Xiaoyu Hu |
| Tsinghua University | | Xiaoyu Hu |

The funders had no role in study design, data collection and interpretation, or the decision to submit the work for publication.

## Author contributions

Tiantian Kou, Data curation, Formal analysis, Validation, Investigation, Visualization, Methodology, Writing – original draft, Writing – review and editing; Lan Kang, Data curation, Formal analysis, Investigation, Visualization, Methodology; Bin Zhang, Software; Jiaqi Li, Investigation; Baohong Zhao, Visualization; Wenwen Zeng, Resources, Methodology; Xiaoyu Hu, Conceptualization, Resources, Data curation, Supervision, Funding acquisition, Validation, Investigation, Visualization, Methodology, Writing – original draft, Project administration, Writing – review and editing

## Author ORCIDs

Lan Kang ⓘ http://orcid.org/0000-0002-8643-9174
Baohong Zhao ⓘ http://orcid.org/0000-0002-1286-0919
Wenwen Zeng ⓘ http://orcid.org/0000-0001-8544-3318
Xiaoyu Hu ⓘ https://orcid.org/0000-0002-4289-6998

## Ethics

This study was performed in strict accordance with the recommendations in the Guide for the Care and Use of Laboratory Animals of the National Institutes of Health. All of the animals were handled according to approved institutional animal care and use committee (IACUC) protocols of the Tsinghua University. The protocol was approved by the Committee on the Ethics of Animal Experiments of the Tsinghua University (17-hxy). All surgery was performed under sodium pentobarbital or avertin anesthesia, and every effort was made to minimize suffering.

Reviewer #1 (Public Review): https://doi.org/10.7554/eLife.88135.3.sa1
Reviewer #2 (Public Review): https://doi.org/10.7554/eLife.88135.3.sa2
Reviewer #3 (Public Review): https://doi.org/10.7554/eLife.88135.3.sa3
Author Response https://doi.org/10.7554/eLife.88135.3.sa4

# Additional files

## Supplementary files

• MDAR checklist

## Data availability

Sequencing data have been deposited in GEO under accession code GSE208772.

The following dataset was generated:

| Author(s) | Year | Dataset title | Dataset URL | Database and Identifier |
|-----------|------|---------------|-------------|-------------------------|
| Kang L, Zhang B, Kou T | 2024 | RNA-seq for blood Ly6Chi and Ly6Clo monocyte in WT and RBP-Jfl/fl Lyz2-Cre mice | https://www.ncbi.nlm.nih.gov/geo/query/acc.cgi?acc=GSE208772 | NCBI Gene Expression Omnibus, GSE208772 |

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

# Appendix 1

**Appendix 1—key resources table**

| Reagent type (species) or resource | Designation | Source or reference | Identifiers | Additional information |
|---|---|---|---|---|
| Strain, strain background (*Mus musculus*) | *Cx3cr1*<sup>gfp/gfp</sup> | Jackson Laboratory | Strain #: 005582 from Jackson Laboratory | |
| Strain, strain background (*M. musculus*) | *Ccr2*<sup>RFP/RFP</sup> | Jackson Laboratory | Strain #: 017586 from Jackson Laboratory | |
| Strain, strain background (*M. musculus*) | *Rbpj*<sup>fl/fl</sup> | Tasuku Honjo of Kyoto University | | |
| Strain, strain background (*M. musculus*) | *Lyz2*-Cre | Jackson Laboratory | Strain #: 004781 from Jackson Laboratory | |
| Strain, strain background (*M. musculus*) | C57BL6/J | Jackson Laboratory | Strain #: 000664 from Jackson Laboratory | |
| Strain, strain background (*M. musculus*) | CD45.1 | Jackson Laboratory | Strain #:002014 from Jackson Laboratory | |
| Antibody | APC/Cy7 anti-mouse CD45 antibody | BioLegend | 103116 | 1:400 |
| Antibody | PE anti-mouse CD45 | BioLegend | 103106 | 1:400 |
| Antibody | Brilliant Violet 510 anti-mouse CD45 antibody | BioLegend | 103137 | 1:400 |
| Antibody | Alexa Fluor 700 anti-mouse Ly-6C | BioLegend | 128024 | 1:400 |
| Antibody | PE anti-mouse Ly-6C | BD Biosciences | 560592 | 1:400 |
| Antibody | CD4 monoclonal antibody, PerCP-Cyanine5.5 | eBioscience | 45-0042-82 | 1:400 |
| Antibody | PE anti-mouse CD3ε antibody | BioLegend | 100307 | 1:400 |
| Antibody | BV605 anti-mouse CD19 antibody | BioLegend | 115539 | 1:400 |
| Antibody | BV421 anti-mouse CD16.2 antibody | BioLegend | 149521 | 1:400 |
| Antibody | CD11c monoclonal antibody, PE-Cyanine7 | eBioscience | 25-0114-82 | 1:400 |
| Antibody | CD11c monoclonal antibody, PerCP-Cyanine5.5 | eBioscience | 45-0114-82 | 1:400 |
| Antibody | Ly-6G monoclonal antibody, APC | eBioscience | 17-9668-82 | 1:400 |
| Antibody | Ly-6G monoclonal antibody, PerCP-eFluor 710 | eBioscience | 46-9668-82 | 1:400 |
| Antibody | CD11b monoclonal antibody, PerCP-Cyanine5.5 | eBioscience | 45-0112-82 | 1:400 |
| Antibody | CD11b monoclonal antibody, PE-Cyanine7 | eBioscience | 25-0112-82 | 1:400 |
| Antibody | CD117 (c-Kit) monoclonal antibody, APC | eBioscience | 17-1171-82 | 1:400 |
| Antibody | CD135 (Flt3) monoclonal antibody, APC, | eBioscience | 17-1351-82 | 1:400 |
| Antibody | CD135 (Flt3) monoclonal antibody, PE | eBioscience | 12-1351-82 | 1:400 |
| Antibody | Brilliant Violet 421 anti-mouse CD192 (CCR2) antibody | BioLegend | 150605 | 1:400 |
| Antibody | Mouse Hematopoietic Lineage Antibody Cocktail, eFluor 450 | eBioscience | 88-7772-72 | 1:400 |
| Antibody | Brilliant Violet 421 anti-mouse CD45.1 antibody | BioLegend | 110732 | 1:400 |
| Antibody | CD45.2 monoclonal antibody, PE-Cyanine7 | eBioscience | 25-0454-80 | 1:400 |
| Antibody | FITC anti-mouse F4/80 antibody | BioLegend | 123107 | 1:400 |
| Antibody | APC/Cyanine7 anti-mouse F4/80 antibody | BioLegend | 123117 | 1:400 |

*Appendix 1 Continued on next page*

*Appendix 1 Continued*

| Reagent type (species) or resource | Designation | Source or reference | Identifiers | Additional information |
|---|---|---|---|---|
| Antibody | APC anti-mouse CD64 antibody | BioLegend | 139306 | 1:400 |
| Antibody | CD206 monoclonal antibody, PE | eBioscience | 12-2061-82 | 1:400 |
| Antibody | PerCP/Cyanine5.5 anti-mouse CD206 antibody | BioLegend | 141715 | 1:400 |
| Antibody | Nur77 monoclonal antibody, PerCP-eFluor 710 | eBioscience | 46-5965-82 | 1:400 |
| Antibody | Ki-67 monoclonal antibody, APC | eBioscience | 17-5698-82 | 1:400 |
| Antibody | CD115 monoclonal antibody, Biotin | eBioscience | 13-1152-85 | 1:400 |
| Antibody | Brilliant Violet 605 Streptavidin | BioLegend | 405229 | 1:400 |
| Antibody | Rabbit anti-GFP antibody | Proteintech | 50430-2-AP | 1:200 |
| Antibody | Goat anti-rabbit Alexa Fluor 488 | Cell Signaling Technology | 4412S | 1:1000 |
| Chemical compound, drug | Phosphate buffered saline (PBS) | Gibco | C10010500BT | |
| Chemical compound, drug | CountBright Absolute Counting Beads | Invitrogen | C36950 | |
| Chemical compound, drug | DAPI | Solarbio | C0060-1ml | |
| Chemical compound, drug | HBSS | MACGENE | CC016.1 | |
| Chemical compound, drug | FBS | Gibco | 16000-044 | |
| Chemical compound, drug | Collagenase type I | Sigma-Aldrich | C0130-500MG | |
| Chemical compound, drug | DNase I | Sigma-Aldrich | 10104159001 | |
| Chemical compound, drug | Percoll | GE Healthcare | 17-0891-01 | |
| Chemical compound, drug | SlowFade Diamond Antifade Mountant | Life Technologies | S36972 | |
| Chemical compound, drug | Tween-20 | Amresco | 0777-1L | |
| Chemical compound, drug | Paraformaldehyde | Sigma-Aldrich | 158127-500G | |
| Chemical compound, drug | OCT | Sakura Finetek | 4583 | |
| Chemical compound, drug | Triton X-100 | Merck Millipore | 648466 | |
| Chemical compound, drug | Glycine | Amresco | 0167-5kg | |
| Chemical compound, drug | Lipopolysaccharide (LPS) | Sigma-Aldrich | L2630 | |
| Commercial assay or kit | Cytofix/Cytoperm Fixation/Permeabilization Solution Kit | BD Biosciences | 554715 | |
| Commercial assay or kit | Click-iT EdU AF488 Flow Cytometry Assay Kit | Invitrogen | C10425 | |
| Commercial assay or kit | Fluoresbrite Polychromatic Red Microspheres | Polysciences | 19507-5 | |
| Commercial assay or kit | 7-AAD Viability Staining Solution | BioLegend | 420403 | |
| Commercial assay or kit | Total RNA Purification Kit | GeneMarkbio | TR01-150 | |
| Commercial assay or kit | FastSYBR mixture | CWBIO | CW2622M | |
| Commercial assay or kit | Reverse Transcriptase M-MLV | Takara | 2641B | |
| Commercial assay or kit | Annexin V Apoptosis Detection Kit APC | eBioscience | 88-8007-72 | |
| Software, algorithm | FlowJo | FlowJo | RRID:SCR_008520 | |
| Software, algorithm | Prism | GraphPad | RRID:SCR_002798 | |
| Software, algorithm | ImageJ | ImageJ | RRID:SCR_003070 | |

