## [Editor Report · eLife assessment]

This study presents a **valuable** examination into the role Notch-RBP-J signaling in regulating monocyte subset homeostasis. The data were collected and analyzed using **solid** and validated methodology and can be used as a starting point for exploring the mechanisms involved in RBP-J signaling in non-classical monocytes. The data presented strongly confirm the authors conclusions. However, this article primarily focuses on providing a description, and additional studies are necessary to fully elucidate the mechanisms through which RBP-J deficiency contributes to the specific increase in Ly6C^lo^ monocyte numbers in both the blood and lungs.

---

## [Referee Report · Reviewer #1 (Public Review)]

Kou and Kang et al. investigated the role of Notch-RBP-J signaling in regulating monocyte homeostasis. Specifically, they examined how a conditional knockout of Rbpj expression in monocytes though a Rbpjfl/fl Lyz2cre/cre mouse affects the homeostasis of Ly6Chi versus Ly6Clo monocytes. They discovered that Rbpj deficiency did not affect the percentage of Ly6Chi monocytes but instead, led to an accumulation of Ly6Clo monocytes in the peripheral blood. Using a comprehensive number of in vivo techniques to investigate the origin of this increase, the authors revealed that the accumulation of Rbpj deficient Ly6Clo monocytes was not due to an increase in bone marrow egress and homing and that this defect was cell intrinsic. However, EdU-labelling and apoptosis assays revealed that this defect was not due to an increase in proliferation nor conversion of Ly6Chi to Ly6Clo monocytes. Interestingly, it was revealed that Rbpj deficient Ly6Clo monocytes had increased expression of CCR2 and ablation of CCR2 expression reversed the accumulation of these cells in the periphery. Lastly, they discovered that Rbpj deficiency also led to downstream effects such as an accumulation of Ly6Clo monocytes in the lung tissue and increased CD16.2+ interstitial macrophages, presumably due to dysregulated monocyte differentiation and function.

Their findings are interesting and highlight a previously unexplored association between Notch-RBP-J signaling and CCR2 expression in monocyte homeostasis, providing further insight into the distinct pathways that regulate Ly6Chi vs Ly6Clo monocyte subsets individually.

The strengths of this paper include the use of multiple conditional genetic knock out mouse models to explore their hypothesis and the use of sophisticated in vivo techniques to study the major mechanisms involved in monocyte homeostasis. However, a major weakness of the paper is the exact role of how CCR2 compensates for the increase in Ly6Clo monocytes in the circulation in the RBP-J knockout mice as the authors showed no differences in their conversion, egress or homing back to the bone marrow. The authors were also unable to show that RBP-J knockout mice were functionally different in their response to CCL2 due to technical difficulties, which makes it challenging to conclude how CCR2 compensates for their trafficking patterns. Consequently the link between CCR2 and RBP-J remains correlative based on the data presented in the paper.

The conclusions of this paper are mostly well substantiated from the experimental data but as mentioned above, the mechanism of how CCR2 relates to the increase in Ly6Clo monocytes in RBP-J knockout mice is still unclear. Nevertheless, this work will be of interest to immunologists and biologists working on Notch-signalling in diseases. In addition, the methods and data would be useful for researchers who are seeking to use the Rbpjfl/fl Lyz2cre/cre mouse model for their studies.

---

## [Referee Report · Reviewer #2 (Public Review)]

The authors provide a compelling data to demonstrate that the Notch-related transcription factor RBP-J can influence the number of circulating and recruited monocytes. The authors first delete the Rbpj gene in the myeloid lineage (Lyz2) and show that, as a proportion, only Ly6Clo monocytes are increased in the blood. The authors then attempted to identify why these cells were increased in proportion but ruled out proliferation or reduced apoptosis. Next, they investigated the gene signature of Rbpj null monocytes using RNA-sequencing and identified elevated Ccr2 as a defining feature. Crossing the Rbpj null mice to Ccr2 null mice showed reduced numbers of Ly6Clo monocytes compared with Rbpj null alone. Finally, the authors identify that an increased burden of blood Ly6Clo monocytes is correlated with increased lung recruitment and expansion of lung interstitial macrophages.

The main conclusion of the authors, that there is a 'cell intrinsic requirement of RBP-J for controlling blood Ly6CloCCR2hi monocytes' is strongly supported by the data. However, other claims and aspects of the study require clarification and further analysis of the data generated.

Strengths

The paper is well written and structured logically. The major strength of this study is the multiple technically challenging methods used to reinforce the main finding (e.g. parabiosis, adoptive transfer). The finding reinforces the fact that we still know little about how immune cell subsets are maintained in situ, and this study opens the way for novel future work. Importantly, the authors have generated an RNA-sequencing dataset that will prove invaluable for identifying the mechanism - they have promised public access to this data via GEO - it is expected this will be made accessible upon publication.

Weaknesses - The main weakness of the study, is that although the main result is solidly supported, as written it is mostly descriptive in nature. For instance, there is no given mechanism by which RBP-J increases Ly6Clo monocytes. The authors conclude this is dependent on CCR2, however CCR2 deletion has a global effect on monocyte numbers and importantly in this study, it does not remove the Ly6Clo bias of cell proportions, if anything it seems to enhance the difference between the ly6C low and high populations in Rbpj null mice (figure 5C). This oversight in data interpretation likely occurred because: (i) this experiment is missing a potentially important control (Lyz2cre/cre Ccr2RFP/RFP or RBP-J variations), and (ii) lack of statistical comparisons between Ly6Clow and high subsets (e.g. two-way ANOVA design). In general, there seemed to be a focus on the Ly6C low cells, where the mechanism may be more identifiable in their precursors - likely the Ly6C high monocytes. Furthermore, the lack of this mechanism and data comparison may also be important, because it is possible that RBP-J signalling merely maintains the expression of Ly6C, rather than controls non-classical monocyte differentiation. In this case the comparison made for the sequencing data would be between Ly6C low non classical monocytes and 'artificially' Ly6C low classical monocytes. The basis of a population based on one marker is currently a widespread flaw in the field.

Other specific weaknesses were identified (note these are in addition to the more important comments above):

1. The confirmation of knockout in supplemental figure 1A shows only a two third knockdown when this should be almost totally gone. The authors have confirmed this is perhaps poor primer design and cite a study which shows almost complete reduction in protein levels (though this could be made more clear).

2. Many figures (e.g. 1A) only show proportional data (%) when the addition of cell numbers would also be informative - for example, what if Ly6Chigh cells were decreasing, thus artificially increasing the proportion of Ly6Clo cells? Looking at figure 7B - where cell numbers are shown, it is clear that cell proportion differences often do not match number data - here RBP-J knockout also increases Ly6C high cells in number (but not %).

3. It was noted previously that many figures only have an n of 1 or 2 (e.g. 2B, 2C), the authors clarified that some of these displayed one dot to represent an experiment of multiple n.

4. There is incomplete analysis (e.g. Network analysis, comparison to subset-restricted gene expression) and interpretation of RNA-sequencing results (figure 4), additionally the difference between the genotypes in both monocyte subsets would provide a more complete picture and potentially reveal mechanisms

5. The experiments in figure 5 are missing a control (Lyz2cre/cre Ccr2RFP/RFP or the Rbpj+/+ versions) and may have been misinterpreted. For example if the control (RBP-J WT, CCR2 KO) was used then it would almost certainly show falling Ly6C low numbers compared to RBP-J WT CCR2 WT, but RBP-J KO CCR2 KO would still have more Ly6c low monocytes than RBP-J WT, CCR2 KO - meaning that the RBP-J function is independent of CCR2. I.e. Ly6c low numbers are mostly dependent on CCR2 but this is irrespective of RBP-J. Explained in another way, the normal ratio of Ly6C high to low is around 1.5 Ly6Chigh cells for every one Ly6Clow cell, this is flipped in the RBP-J knockout to 1 high to 1.25 low (the main finding of the paper), but when CCR2 is removed it actually becomes 1 high to 5 low (actual numbers 0.2% vs around 1%) - in which case all CCR2 removal is doing is lowering the number of monocytes and RBP-J's mechanism is independent of CCR2.

6. Figure 6 was difficult to interpret because of the lack of shown gating strategy. The authors state they copied the strategy from Schyns et al. however in order to review this correctly the authors should show a supplemental figure of their own gating.

7. Figure 7 has the same problem as figure 5, but this time has the correct control. CCR2 ablation has a global suppression of monocyte numbers however the increased ly6c low monocyte ratio is most likely still present in the DKO mice - the lower numbers reduce the clarity of the data. Additionally in Lung IM macrophages depletion of CCR2 in the DKO only had a partial effect in some cell types - so CCR2 is playing a role, but it is not fully dependent. A good comparison would be if they blocked PU.1 expression - the effect of RBP-J would also be muted but it doesn't mean anything in terms of mechanism. Statements about the origin of the cells may need to be removed due to lack of compelling evidence.

8. Even after being notified and acknowledging the study, the authors still have not referred to or cited a similar 2020 study in their manuscript. This study also investigated myeloid deletion of Rbpj (Zhang et al. 2020 - https://doi.org/10.1096/fj.201903086RR). Zhang et al identified that Ly6Clo alveolar macrophages were decreased in this model - it is intriguing to synthesise these two studies and hypothesise that the ly6c low monocytes steal the lung niche, but this was not discussed. The authors also indicated they looked at AM but saw no difference - perhaps they should look specifically at Ly6Clow AMs in their data to compare with this study?

---

## [Referee Report · Reviewer #3 (Public Review)]

In this study, the authors investigate the role of the Notch signalling regulator RBP-J on Ly6Clow monocyte biology starting with the observation that RBP-J-deficient mice have increased circulating Ly6low monocytes. Using myeloid specific conditional mouse models, the authors investigate how RBP-J deficiency effects circulating monocytes and lung interstitial macrophages.

A major strength of this study is that it provides compelling evidence that RBP-J is a novel, critical factor regulating Ly6Clow monocyte cell frequency in the blood. The authors demonstrate that RBP-J deficiency leads to increased Ly6Clow monocytes in the blood and lung and CD16.2+ interstitial macrophages in steady state. The authors use a number of different techniques to confirm this finding including bone marrow transplantation experiments and parabiosis.

The main conclusion of the paper is that RBP-J controls the fate of Ly6ClowCCR2hi monocytes in a cell-intrinsic manner. This conclusion is strongly supported by the data provided. However, this paper is predominantly descriptive and further research is required to fully uncover the mechanisms by which RBP-J deficiency leads to Ly6Clo monocyte numbers increasing specifically in the blood and lungs and the consequence of RBP-J deficiency on Ly6C-low monocyte functionality.

The authors have performed RNA-seq and more in-depth analysis of this sequencing may provide clues for uncovering the thus far elusive mechanism.

---

## [Author Response]

The following is the authors’ response to the original reviews.

We thank the reviewers for their time and insightful and constructive comments. We are pleased that reviewers found this study “opens the way for novel future work” and the findings “interesting”. We have experimentally addressed the points raised by the reviewers and have substantially revised the manuscript by modifying 30 figures panels. The reviewers’ points are specifically addressed below.

1. The authors concluded that an accumulation of Ly6Clo monocytes occurred in the Rbpjfl/fl Lyz2cre/cre mouse by examining the percentage of cells among CD45+ cells in Figure 1. It would be helpful if the authors could give an account of the total cell count numbers of monocyte subsets per ml of blood and in the bone marrow to give the readers a better idea of the extent of increase as cell percentages among CD45+ cells may be influenced by the number of other immune subsets.

We thank the reviewer for raising these points. In this research, we crossed Rbpjfl/fl mice with Lyz2-Cre mice carrying the Cre recombinase inserted in the Lysozyme-M (Lyz2) gene locus results in the selective deletion of RBP-J in myeloid cells, such as monocytes, macrophages and granulocytes. We then proceeded to examine the neutrophil levels in the bone marrow and blood. The percentage of neutrophils observed was found to be similar to that of control mice, which was in line with the findings reported in the literature (Metzemaekers et al. 2020). Furthermore, the proportion of Ly6Chi monocytes in RBP-J deficient mice was found to be similar to that of control mice, which is consistent with the literature (Ginhoux et al. 2014). Based on these results, we thought that the changes observed in the proportion of Ly6Clo monocytes could reliably indicate the alterations occurring in Ly6Clo monocytes within the Rbpjfl/flLyz2cre/cre mice.

1. The authors demonstrated no significant differences in bone marrow progenitor and monocyte numbers, therefore concluding that monocyte egress from the bone marrow did not contribute to the increase in Ly6Clo monocyte numbers in the blood (Figure 1B-D). As it is unclear what is the exact cell number increase in the blood, the changes in bone marrow monocyte numbers might be too small to be reflected in their percentage calculations. In light that CCR2 was also found to play a role in Ly6Clo monocyte homeostasis in Rbpjfl/fl Lyz2cre/cre mice, could the authors demonstrate if Rbpj-deficient Ly6Clo monocytes might be more responsive to CCL2 through transwell experiments? This would also provide readers a more in-depth mechanism of how an increase in CCR2 on Rbpj-deficient Ly6Clo monocytes leads to their accumulation in the periphery.

The experimental results regarding the proportion of monocytes and precursor cells in the bone marrow were derived from multiple experiments. The data obtained from individual experiments as well as the final integrated data did not reveal significant differences between the control mice and Rbpjfl/flLyz2cre/cre mice. Therefore, we believed that even if there were small changes in cell numbers, these differences could still be reflected through alterations in their proportions. We attempted transwell experiments, but unfortunately, they were not technically successful. Nearly all sorted Ly6Clo monocytes attached to the transwell membrane, making it challenging to draw a conclusion regarding the responsiveness of RBP-J deficient Ly6Clo monocytes to CCL2.

1. In the parabiosis experiment conducted in Figure 3C-E, the authors provide conclusive evidence that the accumulation of Rbpj-deficient Ly6Clo monocytes was cell intrinsic as Rbpj-deficient Ly6Clo monocytes continued to accumulate in the blood of control counterparts. Monocytes have also been shown to accumulate in the spleen and re-enter or home back to the bone marrow. Assessing if there is a change in monocyte homing abilities in Rbpj-deficient Ly6Clo monocytes by examining their numbers in the spleen and bone marrow of control parabiotic mice would substantiate their claims that the defect was cell intrinsic and provide further understanding for the readers of why Rbpj-deficient Ly6Clo monocytes accumulate in the blood.

We thank the reviewer for bringing out this interesting point. We also analyzed the proportions of GFP- Ly6Chi monocytes and Ly6Clo monocytes in the bone marrow of parabiotic mice. The experimental results revealed that there were no significant differences in the proportion of GFP- monocytes between the control mice and the KO animals (see the figure A below). We also detected the expression of CXCR4 in bone marrow Ly6Clo monocytes. Rbpjfl/flLyz2cre/cre mice exhibited normal expression of CXCR4 (see Author response image 1 below), which participates in the homing of classical and nonclassical monocytes to bone marrow and spleen monocyte reservoirs (Chong et al. 2016). The homing abilities of RBP-J deficient Ly6Clo monocytes may not have changed.

**Author response image 1. sa4fig1:** 

1. Authors should provide cell counts for Figure 5B to demonstrate the extent CCR2 depletion affects the number of Ly6Clo monocytes in Rbpjfl/fl Lyz2cre/cre mice as explained in point 1.

As mentioned before, we believed that the proportion of circulating monocytes could, to some extent, provide evidence of the impact of CCR2 deficiency on Ly6Clo monocytes.

**Reviewer #2**
1. The confirmation of knockout in supplemental figure 1A shows only a two third knockdown when this should be almost totally gone. Perhaps poor primer design, cell sorting error or low Cre penetrance is to blame, but this is below the standard one would expect from a knockout.

Kang et al (PMID: 31944217) evaluated the knockout efficiency of Rbpj in sorted colonic macrophages of Rbp-jfl/flLyz2cre/cre mice using qPCR and immunoblotting. The qPCR result indicated a two-third knockdown, while the immunoblotting results demonstrated efficient deletion of RBP-J protein in Rbp-jfl/flLyz2cre/cre mice. As pointed out by the reviewer, the observed two-third knockdown, which is lower than the expected complete knockout, may be attributed to primer design.

1. Many figures (e.g. 1A) only show proportional data (%) when the addition of cell numbers would also be informative

We appreciate the reviewer for bringing up these points. Indeed, multiple articles studying monocytes only show changes in cell proportions. As mentioned above, we believed that analyzing the proportion of circulating monocytes could offer valuable evidence of the influence of RBP-J deficiency on Ly6Clo monocytes.

1. Many figures only have an n of 1 or 2 (e.g. 2B, 2C)

Here, we employed annexin V (AnnV) and propidium iodide (PI) staining to evaluate apoptosis and cell death in Ly6Chi and Ly6Clo blood monocytes from control and RBPJ deficient mice. The results showed no significant difference in the levels of apoptosis and cell death between the two groups (see Author response image 2 below). The statistical data for Ki-67 expression obtained from multiple experiments, and the expression of Ki-67 showed no significant difference between the control and RBP-J deficient mice (see the figure B below). In Figure 2C, each dot represents 2-3 mice, and there were no differences observed between control and RBP-J deficient mice at multiple time points during the repeated measurements.

**Author response image 2. sa4fig2:** 

1. Sometimes strong statements were based on the lack of statistical significance, when more n number could have changed the interpretation (e.g. 2G, 3E)

We have derived the corresponding conclusions based on the observed experimental results.

1. There is incomplete analysis (e.g. Network analysis) and interpretation of RNAsequencing results (figure 4), the difference between the genotypes in both monocyte subsets would provide a more complete picture and potentially reveal mechanisms

We thank the reviewer for bringing out this point. We agreed that a more comprehensive analysis, including a comparison between the genotypes in both monocyte subsets, would provide a deeper understanding and potentially uncover underlying mechanisms. Having observed alterations in blood Ly6Clo monocytes in RBP-J deficient mice, our primary focus had been on analyzing the differentially expressed genes within this subset of monocytes to gain further insights into its specific characteristics and behavior. We also uploaded sequencing data sets in the Genome Expression Omnibus with assigned accession numbers GSE208772 to facilitate interested researchers in accessing and downloading the data.

1. The experiments in Figures 5 and 7 are missing a control (Lyz2cre/cre Ccr2RFP/RFP or the Rbpj+/+ versions) and may have been misinterpreted. For example if the control (RBP-J WT, CCR2 KO) was used then it would almost certainly show falling Ly6C low numbers compared to RBP-J WT CCR2 WT, but RBP-J KO CCR2 KO would still have more Ly6c low monocytes than RBP-J WT, CCR2 KO - meaning that the RBP-J function is independent of CCR2. I.e. Ly6c low numbers are mostly dependent on CCR2 but this is irrespective of RBP-J.

The diminished Ly6Clo monocytes in Rbpjfl/flLyz2cre/creCcr2RFP/RFP (DKO) mice can be divided into two distinct subpopulations: one portion originates from Ly6Chi monocytes, while the other comprises Ly6Clo monocytes characterized by heightened CCR2 expression. The Ly6Clo monocytes that remain in DKO mice exhibit CCR2 expression levels within the normal range when compared to Lyz2cre/cre mice, but lower levels compared to RBP-J deficient mice (Figure 5A). These findings suggest that RBP-J exerts regulatory influence over Ly6Clo monocytes, at least in part, through CCR2.

1. Figure 6 was difficult to interpret because of the lack of shown gating strategy. This reviewer assumes that alveolar macrophages were gated out of analysis

The gating strategy of lung interstitial macrophage in the manuscript Figure 6 was consistent with the published work (Schyns et al, cited in the manuscript). We also measured alveolar macrophages (AM) from control and RBP-J deficient mice bronchoalveolar lavage fluid. At the resting state, RBP-J deficient mice exhibited normal AM frequency and number (see Author response image 3 below).

**Author response image 3. sa4fig3:** 

1. The statements around Figure 7 are not completely supported by the evidence, (i) a significant proportion of CD16.2+ cells were CCR2 independent and therefore potentially not all recently derived from monocytes, and (ii) there is nothing to suggest that the source was not Ly6C high monocytes that differentiated - the manuscript in general seems to miss the point that the source of the Ly6C low cells is almost certainly the Ly6C high monocytes - which further emphasises the importance of both cells in the sequencing analysis

Schyns et al and Sabatel at al showed that the numbers of IM and CD16.2+ were similar in Ccr2 sufficient and Ccr2-/- mice, demonstrating that CD16.2+ cells were Ccr2 independent. The number of CD16.2+ cells was significantly reduced in Rbpjfl/flLyz2cre/creCcr2RFP/RFP mice as compared to Rbpjfl/flLyz2cre/cre mice, in line with decreased number of lung Ly6Clo monocytes and blood Ly6Clo monocytes, showing that CD16.2+ cells depended on Ccr2 for their presence in Rbpjfl/flLyz2cre/cre mice.

1. The authors did not refer to or cite a similar 2020 study that also investigated myeloid deletion of Rbpj (Qin et al. 2020 - https://doi.org/10.1096/fj.201903086RR). Qin et al identified that Ly6Clo alveolar macrophages were decreased in this model - it is intriguing to synthesise these two studies and hypothesise that the ly6c low monocytes steal the lung niche, but this was not discussed

We thank the reviewer for bringing this study to our attention. According to their findings, myeloid-specific RBP-J deficiency resulted in a decrease in Ly6CloCD11bhi alveolar macrophages but an increase in Ly6CloCD11blo alveolar macrophages after bleomycin treatment, while the total number of alveolar macrophages showed no significant difference. These results suggest that RBP-J may play a role in regulating the balance between these specific alveolar macrophage subsets in response to bleomycin-induced injury, without affecting the overall population of alveolar macrophages. This may be different from what we observe in interstitial macrophages under resting conditions.

**Reviewer #3**
1. It is curious that the authors do not see the increase in circulating monocytes reflected in the spleen however, the n-number is 2. Increasing the n-number would enable the author to understand the data which is not interpretable at the moment. There are multiple other places in which a low n-number makes it hard to fully understand the biology (eg Figure 2C&E)

Although we only counted the number of splenic monocyte subsets in two mice, the proportion of splenic monocyte subsets was calculated based on additional quantity of mice in our study.

1. Given that Ly6Clow monocytes are thought to be longer lived than Ly6C+ and there is still considerable labelling of Ly6Clow monocytes at the end of the 96 hours analysed in the EdU experiment, it is not possible to determine from the data here whether RBPJ deficiency increases life span. Could it be that differences in %EdU+ cells would only be seen at later time points? If the timeline was extended, could it be that differences in %EdU+ become apparent

Based on the latex bead experiment, we observed that the presence of latex+ Ly6Clo monocytes at 7 days in control and RBP-J deficient mice did not differ, indicating that the lifespan of Ly6Clo monocytes did not increase.

1. Similarly for the latex bead experiment. Given that there is only n=2 at the first time point and only ~30% of Ly6Clow monocytes are Latex+, it is very hard to conclusively claim that RBP-J does not influence monocyte survival or proliferation. An interesting experiment to assess whether RBP-J is increasing monocyte survival could be an adoptive transfer model in which Ly6Clow monocytes are injected into a congenic mouse and tracked over time.

In RBP-J deficient mice, there was an increase in the proportion of Ly6Clo monocytes. We hypothesized that this lower proportion of latex+ cells might make it easier to observe differences, but clearly, in our experiment, no differences were observed between control and RBP-J deficient mice.

1. RNA-seq: Ccr2 and Itgax are not the top hits. The authors do not investigate the top hits which may provide very interesting insight into how RBP-J influences monocyte biology.

We thank the reviewer for raising these points. We also analyzed some top changed genes. The top two gene in the downregulated gene list are Hes1 and Nrarp, which are regulated by the Notch pathway (Krebs et al 2001 and Radtke et al 2010). We tested blood monocytes, but the population of monocyte subsets displayed no differences between Hes1fl/flRbp-jfl/flLyz2cre/cre and Rbp-jfl/flLyz2cre/cre mice (data not shown). As shown in Figure 2- figure supplement 1A, expression of Nr4a1 showed no significant differences between control and RBP-J deficient mice. The top gene in the upregulated gene list is Erdr1, which has been reported to play a role in cellular survival (Soto et al 2017), while blood monocyte subsets in RBP-J deficient mice displayed normal survival.

1. The PCA plot in figure 4C- it would be interesting to see where all the biological replicates fall.

We agree with the reviewer’s assessment that observing the positions of all biological replicates on the PCA plot may indeed yield valuable insights. However, it is worth noting that the upregulated and downregulated genes also offer suggestive hints.

1. Based on CCR2 expression and CD11c expression, monocytes from RBP-J deficient mice look more like Ly6C+ monocytes - could it be that RBP-J is increasing conversion from Ly6C+ monocytes to Ly6Clow? Or could it be that Ly6Clow monocytes are heterogeneous and RBP-J is increasing survival or conversion of one subtype of Ly6Clow monocytes but looking at all Ly6Clow monocytes together is masking this?

Ly6Clo monocyte can be subdivided into different subpopulations depending on surface makers, such as CD43, MHC-II, CD11c and CCR2 (Jakubzick et al 2013 and Ginhoux et al. 2014). Carlin et al founded that a subset of blood Ly6Clow cells was independent of both Ccr2 and Nr4a1. As said by the reviewer, Ly6Clo monocytes are heterogeneous. Therefore, there is a possibility of altered survival in a certain group of Ly6Clo monocytes.

1. The data presented here suggest that lung CD16.2+ interstitial macrophages are derived from Ly6Clow monocytes which are increased via CCR2. Although the data are suggestive, they are not conclusive, lineage tracing and CCR2 blockade or better, conditional CCR2 deficiency would help to strengthen the claim.

Schyns et al showed that the number of CD16.2+ was similar in Ccr2 sufficient and Ccr2-/- mice, demonstrating that CD16.2+ cells were Ccr2 independent. While number of CD16.2+ cells was significantly reduced in Rbpjfl/flLyz2cre/creCcr2RFP/RFP mice as compared to Rbpjfl/flLyz2cre/cre mice, in line with decreased number of lung Ly6Clo monocytes and blood Ly6Clo monocytes. Moreover, the turnover of lung Ly6Chi and Ly6Clo monocytes was normal. These results implicated that CD16.2+ cells depended on Ccr2 for their presence in Rbpjfl/flLyz2cre/cre mice.

1. The figures could do with more headings/ more detailed legends to help the reader, for example including what is BM, what is blood, what is spleen. Figure 2E needs the days labelled on or above the histograms.

We thank the reviewer for raising this important point. We have now added additional detailed legends to the figure.

1. Gating strategies should be included to help the reader understand which cells you are looking at, especially for Figure 6&7.

The gating strategy for Figures 6 and 7 followed the method reported in the literature, which included the identification of alveolar macrophages. Additionally, we labeled the markers for cell populations in the figure.